



# Combining satellite data and appropriate objective functions for improved spatial pattern performance of a distributed hydrologic model

Mehmet C. Demirel[1], Juliane Mai[2,4], Gorka Mendiguren[1], Julian Koch[1,3], Luis Samaniego[2], Simon Stisen[1]

[1]Geological Survey of Denmark and Greenland, Øster Voldgade 10, 1350 Copenhagen, Denmark

[2]Department Computational Hydrosystems, UFZ—Helmholtz Centre for Environmental Research, Leipzig

[3]Department of Geosciences and Natural Resource Management, University of Copenhagen, Copenhagen, Denmark

[4]Department of Civil and Environmental Engineering, University of Waterloo, Waterloo, Canada

*Correspondence to*: Simon Stisen (sst@geus.dk)

**Abstract.** Satellite based earth observations offer great opportunities to improve spatial model predictions by means of spatial pattern oriented model evaluations. In this study, observed spatial patterns of actual evapotranspiration (AET) are utilized for spatial model calibration tailored to target the pattern performance of the model. The proposed calibration framework combines temporally aggregated observed spatial patterns with a new spatial performance metric and a flexible spatial parameterisation scheme. The mesoscale Hydrologic Model (mHM) is used to simulate streamflow and AET and has been selected due to its soil parameter distribution approach based on pedo-transfer functions and the build in multiscale parameter regionalization. In addition two new domain specific spatial parameter distribution options have been incorporated in the model in order to increase the flexibility of root fraction coefficient and potential evapotranspiration correction parameterisations, based on soil type and vegetation density. These parametrisations are utilized as they are most relevant for simulated AET patterns from the hydrologic model. Due to the fundamental challenges encountered when evaluating spatial pattern performance using standard metrics, we developed a simple but highly discriminative spatial metric i.e. comprised of three easily interpretable components measuring co-location, variation and distribution of the spatial data.

The study shows that with flexible spatial model parameterisation used in combination with the appropriate objective functions, the simulated spatial patterns of actual evapotranspiration become substantially more similar to the satellite based estimates. Overall 26 parameters are identified for calibration through a sequential screening approach based on a combination of streamflow and spatial pattern metrics. The robustness of the calibrations is tested using an ensemble of nine calibrations based on different seed numbers using the shuffled complex evolution optimizer. The calibration results reveal a limited trade-offs between streamflow dynamics and spatial patterns illustrating the benefit of combining separate observation types and objective functions. At the same time, the simulated spatial patterns of AET significantly improved when including an objective function based on observed AET patterns and a novel spatial performance metric compared to traditional streamflow only calibration. Since the overall water balance is usually a crucial goal in the hydrologic modelling, spatial pattern oriented optimization should always be accompanied by traditional discharge measurements. In such a multi-objective framework, the current study promotes the use of a novel bias-insensitive spatial pattern metric, which exploits the key information contained in the observed patterns while allowing the water balance to be informed by discharge observations.





## 1 Introduction

Reliable estimations of spatially-distributed actual evapotranspiration (AET) are useful for various sustainable water resources management practices such as irrigation planning, agricultural drought monitoring and water demand forecasting in large cultivated areas (Wei et al., 2017). Distributed hydrologic models can potentially provide this insight since ET is a

major part of the water cycle. In spite of their ability to simulate detailed spatial patterns of a range of hydrological state variable and fluxes, distributed model evaluation remains focused on temporal aspects of the aggregated streamflow variable (Demirel et al., 2013; Schumann et al., 2013). We are interested in including spatial AET patterns in the model calibration using spatial parameterisations and complementary objective functions. Different methods exist that utilize satellite based land surface temperature data to derive spatially detailed estimates of latent heat fluxes from land-surface and canopy at a

scale relevant for catchment modelling (Kalma et al., 2008). Since AET cannot be measured directly by satellite, surface energy balance models are developed to estimate AET based on data from a range of spectral and thermal bands (Guzinski et al., 2013; Norman et al., 1995; Su, 2002). While these satellite-based estimates are usually employed as a tool to understand and improve the model parameterisations (Conradt et al., 2013; Hunink et al., 2017; Schuurmans et al., 2011), they can also be used to calibrate models (Crow et al., 2003; Immerzeel and Droogers, 2008; Zhang et al., 2009). Therefore, adding

satellite based observations to model calibration is not novel; however, specifically evaluating spatial patterns in the calibration has been rarely done (Stisen et al., 2011b). Interesting examples exist where model calibration could benefit from the spatial pattern information of actual evapotranspiration (Githui et al., 2016; Li et al., 2009; Zhang et al., 2009) and satellite based recharge patterns (Hendricks Franssen et al., 2008). This paper utilizes monthly patterns of AET first to understand and organize ET related model spatial parameterisations then to pursue a calibration. This is from the fact that

adding only temporal aspect of the spatial observations to the objective function is not sufficient for achieving significant improvements in simulated spatial patterns if model parameterisation is not flexible enough to physically adjust to the observed pattern. Besides, the model structure, parameterisations and calibration schemes have usually been designed for streamflow optimizations (Vazquez et al., 2011; Velázquez et al., 2010). In order to ensure compatibility between the spatial pattern calibration target and model parameterisation, the flexibility of the spatial model parameterisation needs to be

reconsidered. Recently, inadequate representation of spatial variability and hydrologic connectivity of a well-known distributed model (VIC) has been reported by Melsen et al. (2016). The mesoscale Hydrologic Model (mHM) has the flexibility to alter the spatial patterns via pedo-transfer function (PTF) parameters and by including a multi-scale parameter regionalization (MPR) scheme (Rakovec et al., 2016; Samaniego et al., 2010). Mizukami et al. (2017) incorporated this MPR approach with VIC to estimate parameters for large domain based on geophysical data for 531 basins. The multi-basin

calibration results using MPR revealed physically meaningful parameter fields without patchiness (discontinuities). The study by Loosvelt (2013) is one of few other examples that incorporate PTFs for soil texture and moisture components of a hydrologic model.

All calibration strategies rely on the selection of performance metrics indicating the goodness-of-fit of the model to be optimized. Choosing an appropriate set of objective functions is crucial to build a robust calibration strategy since there will

be trade-offs between different objective functions or redundant information. In the hydrology literature, there are a range of different temporal metrics for hydrograph match while metrics designed for spatial pattern matching are less common (Koch et al., 2017b; Rees, 2008). For distributed models, spatial metrics usually evaluate cell-to-cell correlation and deviations (e.g. Pearson's R and bias). The use of multi-component metrics as described for discharge by Gupta et al. (2009) is however rare for spatial pattern evaluation. An essential feature of our study is introducing a new spatial efficiency (SPAEF) metric that

contains three components i.e. correlation, variance and histogram intersection providing reliable bias-insensitive pattern information unlike other traditional metrics focusing on only one aspect like correlation, mean squared error or bias.

Prior to model calibration, sensitivity analysis is usually conducted to attribute response of the model outputs to the changes in model parameters (Shin et al., 2013), which can enhance our understanding of both temporal and spatial model behaviour





(Berezowski et al., 2015). In the context of spatial model calibration, the sensitivity analysis should not only identify the parameters that affect the water balance and hydrograph dynamics but also the parameters that shape the spatial patterns of the simulated states and fluxes. To achieve this, we have to design objective functions that reflect the spatial pattern of the models and utilize these in model parameter sensitivity analysis.

In light of the well-known equifinality problems in model calibration (Beven and Freer, 2001) spatial pattern evaluation can be useful for selecting the most appropriate parameter set from a group of sets leading to both reasonable streamflow performance and physically meaningful AET pattern. Immerzeel and Droogers (2008) showed that the models can be constrained by using spatially distributed observations with a monthly temporal resolution. Cornelissen et al. (2016) highlighted the need to identify which model parameters influence the simulated spatial pattern and showed that spatial

patterns of simulated evapotranspiration were most sensitive to the land-use parameterisation, whereas precipitation was the most sensitive input data with respect to temporal dynamics of the model. Rakovec et al. (2016) used total water storage (TWS) anomaly from the Gravity Recovery and Climate Experiment (GRACE) satellites and evapotranspiration estimates from FLUXNET data (https://fluxnet.ornl.gov/) to improve model parameterisations for discharge simulations. They showed that adding TWS anomalies to the calibration performed reasonably well for continental 83 European basins with

different climatology.

   The main objectives of this study are to 1) determine appropriate spatial model parameterisations and objective functions for the domain and 2) incorporate spatial patterns of satellite based actual evapotranspiration data in the model calibration and validation. For that we investigate the ET related model parameterisations and evaluate both temporal dynamics of streamflow and spatial patterns of AET over the basin. In order to improve AET simulations, we use transfer functions that

combine a priori maps of soil and vegetation properties with few global calibration parameters in order to enhance the spatial parameterisation flexibility and allow the parameter field to adjust to an observed spatial patterns of AET from the catchment. We also design a new multi-component metric specifically suited for comparing two spatial patterns of continuous variables. This requires bias insensitive components evaluating different crucial characteristics of the spatial data. Here, we prioritize three main data properties, which are co-location, variation and distribution. In addition to the streamflow

observations we use remote sensing AET data to calibrate and evaluate the distributed model for the second research objective. Prior to calibration, important parameters are identified using a sensitivity analysis including both streamflow dynamics and spatial patterns of AET as objective functions. We then use a well-known global search algorithm of SCE-UA (Duan et al., 1992) to calibrate the mHM hydrologic model. The calibration is conducted using three strategies for objective function selection. First, streamflow metrics and spatial pattern metrics are used in isolation during calibration and

subsequently they are combined in a more balanced model optimization. In this way we can investigate the trade-offs and robustness of the different approaches by evaluation the performances regarding both streamflow and spatial patterns during calibration and validation.

## 2 Study Area and Data

### 2.1 Study area

The Skjern river basin is one of the most popular research basins in Denmark as it is highly instrumented for hydrological monitoring including eddy-flux towers, a dense soil moisture network and other state-of-the-art monitoring of hydrological variables (Jensen and Illangasekare, 2011). The basin area is approximately 2500 km$^2$ containing mostly sandy soils (Figure 1). The river is the largest in Denmark by flow volume and located in the western part of the Jutland peninsula, a region dominated by agriculture and forests together covering ~80% of the domain (Larsen et al., 2016). The basin is mostly flat

with a maximum altitude of 130 m and it receives a mean annual precipitation of around 1000 mm (Stisen et al., 2011a). The mean annual streamflow is around 475 mm and monthly mean temperatures varies from 2 up to 17°C (Jensen and



Illangasekare, 2011).

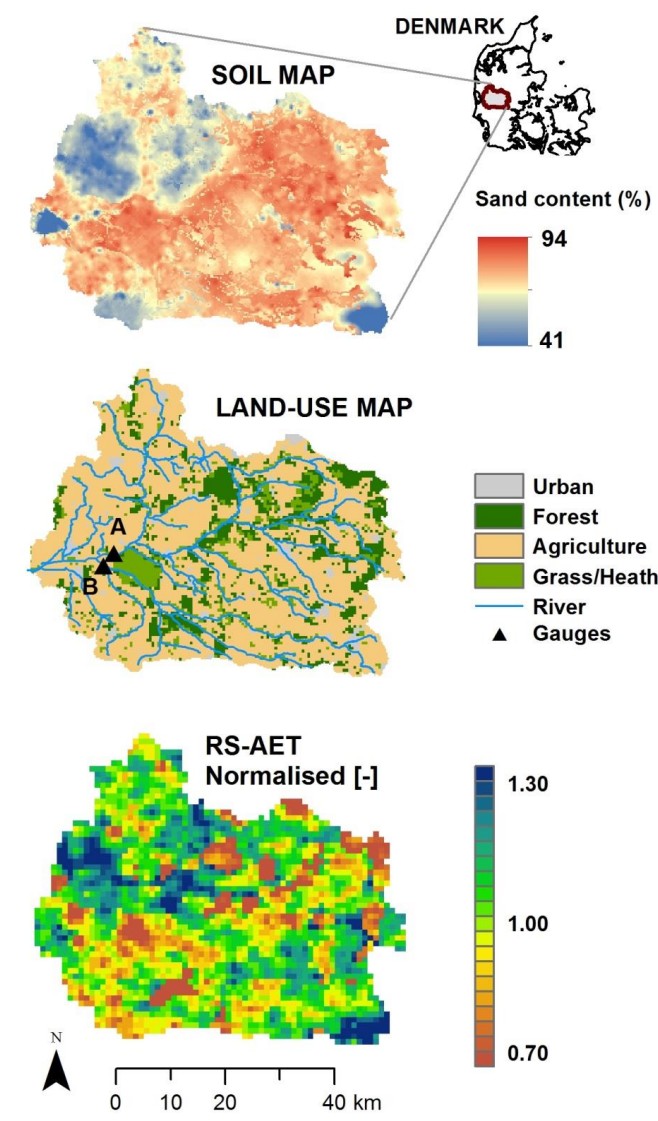

**Figure 1 Skjern River Basin location, soil type and land-use characteristics. An average pattern of satellite based actual evapotranspiration for June (average of all years from 2001 until 2008) is also presented to illustrate the interaction between soil type and land-use that generate the land surface flux patterns.**

**2.2 Satellite based data**

The Moderate Resolution Imaging Spectroradiometer (MODIS), polar orbiting platforms, Terra and Aqua, observe mid-latitude regions four times per day at a spatial resolution of approximately 1x1km. The Two Source Energy Balance (TSEB) model proposed by Norman et al. (Norman et al., 1995) based on the Priestly-Taylor approximation (Priestley and Taylor, 1972) is used in this study to calculate AET based on MODIS data under cloud-free conditions. The model inputs are land surface temperature (LST), solar zenith angle (SZA), as well as albedo and height of canopy all derived from MODIS observations (Mendiguren et al., 2017). Additional inputs such as climate variables of air temperature and incoming



radiation are obtained from ERA interim reanalysis data (Dee et al., 2011). The main motivation of preparing a new AET dataset based on land surface temperature is that most other available products are based mainly on vegetation index data which may not be sufficient to assess the complicated interplay between climate, soil and vegetation dynamics on the AET patterns especially during the growing season. For more details of our newly produced AET data for Denmark, including

equations, calibration and validation, please refer to the recent study by Mendiguren et al. (2017).

In this study, all remote sensing based AET data were averaged for each month s during the growing season across all years for the model calibration period (2001-2008) to six monthly mean maps from April to September under cloud free condition. This ensures that in spite of uncertainty in the individual instantaneous midday estimates of AET, the monthly maps represent the general spatial pattern for each month under cloud-free conditions. Despite not being pure observations but

estimates from an energy balance model based on satellite observations we will refer to these AET maps as reference observation.

## 3 Hydrologic Model

The mesoscale Hydrologic Model (mHM) is a distributed model providing various simulated spatial outputs, fluxes and states at different spatiotemporal model resolutions (Samaniego et al., 2010). The model includes pedo-transfer functions for

soil parameterisation and originally contains 53 global parameters that can be adjusted during calibration. In this study, some parameters are fixed at a default value and others have been added from the new spatial model parametrizations resulting in a total of 48 global parameters for further analysis. The model simulates major components of the hydrologic cycle i.e. interception, infiltration, snow accumulation and melting, evapotranspiration, groundwater storage, seepage and runoff generation. The readers are referred to the study by Samaniego et al. (2010) for full model description, assumptions,

limitations and process formulations.

Table 1 provides a summary of the modelling data used in this study. As shown in the table, meteorological data can be at a different spatial scale than both morphological data and the model scale. This flexibility arises from the fact that mHM incorporates a multi-parameter regionalization technique to swap between different scales while calculating all fluxes and routing streamflow at a preferred model scale. We run the model at 1x1 km spatial scale and daily time step. Some processes

like ET are calculated at hourly time step then the final results are aggregated to daily values. All morphological data are prepared at 250x250 m scale. All three meteorological data sets, i.e. P, $ET_{ref}$ and $T_{avg}$, were originally at 10-20 km resolution. We resampled them to 1x1 km using cubic interpolation. This interpolation method is used to avoid patchiness in model simulations due to coarse grids at the native scale of the metrological data. We use 12 monthly LAI maps to represent the climatology for both interception and PET correction for the entire period (2001-2014) and the model warm-up period

(1997-2000). There is a disaggregation module in mHM model transforming 12 LAI monthly values to daily values which accelerates the model runs.

**Table 1 Overview of morphological and meteorological data used as input for mHM. Acronyms: BIOS: BioScience Aarhus University, DMI: Danish Meteorological Institute, GEUS: Geological Survey of Denmark and Greenland, MODIS: Moderate**

**Resolution Imaging Spectroradiometer, DGA: Danish Geodata Agency**

| Variable | Description | Warm-up Period | Calibration Period | Validation Period | Spatial Resolution | Source |
|---|---|---|---|---|---|---|
| Q (daily) | Streamflow | 1997-2000 | 2001-08 | 2009-14 | Point | BIOS |
| P (daily) | Precipitation | 1997-2000 | 2001-08 | 2009-14 | 10 km | DMI |
| $ET_{ref}$ | Reference | 1997-2000 | 2001-08 | 2009-14 | 20 km | GEUS and |





| | | | | | | |
|---|---|---|---|---|---|---|
| (daily) | evapotranspiration | | | | | DMI |
| T$_{avg}$ (daily) | Average air temperature | 1997-2000 | 2001-08 | 2009-14 | 20 km | GEUS and DMI |
| LAI | Fully distributed 12 monthly values based on 8-day time varying Leaf Area Index (LAI) dataset | 12 monthly values | 12 monthly values | 12 monthly values | 1 km | MODIS and Mendiguren *et al* (2017). |
| Land cover | Forest, pervious and urban | - | - | - | 250 m | GEUS |
| DEM related data | Slope, aspect, flow accumulation and direction | - | - | - | 250 m | DGA |
| Geology class | Two main geological formations | - | - | - | 250 m | GEUS |
| Soil class | Fully distributed soil texture data | - | - | - | 250 m | Greve et al. (2007) |

### 3.1 Spatial model parameterisation

In order to facilitate a meaningful spatial pattern oriented calibration of a distributed model, we need to compromise between comprehensive (each cell in the basin) and lumped (one cell – one basin) parametrizations as the first approach may require an immense computer resource during calibration and the latter approach usually results in a uniform pattern. For instance, in

a detailed calibration study by Corbari et al. (2013), each pixel in the catchment is represented by a parameter whereas, in a coarse parametrization, a uniform parameter represents the entire catchment (Stisen et al., 2017). In this study, we follow an intermediate level of parametrization comprised of several flexible spatial parameters and nonlinear equations allowing us to stretch the spatial contrast based on soil and vegetation in the catchment. This level of parametrization is still physically meaningful as the parameters are tied to the land surface characteristics of the basin via transfer functions.

**Distributed root fraction coefficient**

Following the domain-specific geophysical literature (Jensen et al., 2001; Madsen and Platou, 1983), we define the root fraction coefficient as a function of field capacity (FC). Similarly following the concept of crop coefficient (Allen et al., 1998), we define the PET correction factor as a function of LAI to represent the vegetation more realistically than a uniform correction factor based on the aspect ratio. These two new spatial model parametrizations are used to increase the realization

capability of the model by increasing the model freedom. Here, spatial model parameterisation is implemented to the root fraction calculation in the original mHM structure which follows the asymptotic equation for vertical root distribution (Eq. (1)) proposed by Jackson et al (1996).

$$Y = 1 - (\beta_c)^d \tag{1}$$

Where $Y$ is the cumulative root fraction from soil surface to depth $d$ (cm), and $\beta_c$ is the root fraction coefficient. We

substituted the root fraction coefficient for pervious areas (non-forest) with two new root fraction parameters i.e. one root fraction for maximum FC (clay) and one for minimum FC (sand) which allow for full spatial distribution of root fraction with varying FC. This relation between soil characteristics and effective rooting depth is based on a domain-specific database with more than 100 soil and root profiles collected in Denmark (Table 19.4 in Jensen et al., 2001) and the literature focusing on soil texture and effective rooting depths in Denmark (Madsen, 1985, 1986; Madsen and Platou, 1983). The




approach is not necessarily globally valid, but designed to the specific region of Western Denmark where very sandy soils (Figure 1) are cultivated for agricultural purposes even though the soil properties impede root development. The two newly introduced parameters are more effective when fine vertical discretization of soil layers is applied. These parameters are used to form the root fraction coefficient for pervious soil ($\beta_{\text{pervious}}$) based on field capacity dependent root fraction at Eq. (2) and (3).

$$FC_{norm} = \frac{FC_i - FC_{min}}{FC_{max} - FC_{min}} \tag{2}$$

where $FC_{norm}$ is the normalized field capacity ranging from 0 to 1.

$$\beta_{pervious} = (FC_{norm} * \beta_{max}) + (1 - FC_{norm}) * \beta_{min} \tag{3}$$

where $\beta_{pervious}$ is the new root fraction for pervious soil comprised of root fraction for clay ($\beta_{max}$) and root fraction for sand ($\beta_{min}$).

**Dynamic ET$_{\text{ref}}$ scaling function**

As a second spatial parametrization step, we incorporated remotely sensed vegetation information, to downscale coarse climatological reference evapotranspiration (ET$_{\text{ref}}$) to the model scale. This was done to emphasize the effect of vegetation on the simulated spatial patterns of AET. The original scaling factor in mHM is based on a lumped minimum correction and an aspect driven additional term. Using aspect ratio for ET$_{\text{ref}}$ correction makes sense in mountainous areas; however this is found irrelevant for the Skjern basin which is characterized by a low topographical variation. The dynamic scaling function introduced here allows the modeller to superimpose the imprint of LAI on the simulated AET patterns via a downscaling of the ET$_{\text{ref}}$. The concept of dynamic scaling function (DSF), given in equation (4), is similar to the concept of crop coefficient as our implementation follows the equation for natural vegetation originally proposed by Allen et al (1998). Similarly, Hunink et al. (2017) compared different applications of crop coefficient in hydrologic modelling. They found that the effect of crop coefficient parameterisations on the water balance is trivial and constant throughout the year; however, it has a major effect on seasonal evapotranspiration and soil moisture fluxes showing the crucial role of crop coefficient for spatial calibration. The DSF has three dimensionless parameters that can be calibrated and a spatio-temporal LAI (-) component to account for the effects of characteristics that distinguish actual vegetation from reference grass (well-watered 10 cm height and having albedo of 0.23). These characteristics include specific land cover, albedo and aerodynamic resistance (Allen et al., 1998; Liu et al., 2017). This ensures a physically meaningful downscaling from a coarse (here 20 km) ET$_{\text{ref}}$ grid to the model resolution (here 1 km).

$$DSF = a + b\left(1 - e^{(-c \cdot LAI)}\right) \tag{4}$$

where $a$ in the model (ET$_{\text{ref}}$-$a$) is the intercept term representing uniform scaling, $b$ (ET$_{\text{ref}}$-$b$) represents the vegetation dependent component while $c$ (ET$_{\text{ref}}$-$c$) describes the degree of nonlinearity in the LAI dependency.

**4 Methods**

In this study, we applied a recently developed sequential screening method (Cuntz et al., 2015) to select important parameters for calibration. Since different parameters can be sensitive to different hydrologic processes, we tested three different performance metrics to evaluate process-parameter relationships. Two of these metrics are derived from the hydrograph i.e. Kling-Gupta Efficiency (KGE, Gupta et al. (2009)) and KGE of only below average streamflow (KGE$_{\text{low}}$) whereas the spatial efficiency metric focuses on the spatial pattern of actual evapotranspiration.



### 4.1 Objective functions

As an objective function for streamflow performance, we chose the Kling–Gupta efficiency, shown at Eq. (5), (KGE; Kling and Gupta, 2009) and applied it to both the entire time series and to the low flow part of the hydrograph (below mean discharge).

$$KGE = 1 - \sqrt{\left(\alpha_Q - 1\right)^2 + \left(\beta_Q - 1\right)^2 + \left(\gamma_Q - 1\right)^2} \tag{5}$$

$$\alpha_Q = \rho(S, O) \text{ and } \beta_Q = {}^{\sigma_O}/_{\sigma_S} \text{ and } \gamma_Q = \frac{(\mu_S - \mu_O)}{\sigma_O}$$

where $\alpha_Q$ is the Pearson correlation coefficient between observed and simulated discharge time series, $\beta_Q$ is the relative variability based on the fraction of standard deviation in simulated and in observed values and $\gamma_Q$ is the bias term normalized by the standard deviation in the observed data. Since comparison of two spatial pattern maps is of obvious importance, a bias-insensitive spatial performance metric is developed and used in this study.

In this context, we adopted the structure of the Kling–Gupta efficiency while substituting the standard deviation term by a

term based on the coefficient of variation $\left({}^{\sigma_O}/_{\sigma_S}\right)$ and replacing the bias term with a histogram comparison index to compare the intersection-percentage of two histograms i.e. observed and simulated histograms. The main utility of the histogram comparison is that it distinguishes between different soil and vegetation groups reflected in the spatial pattern results. This unique feature of being sensitive to clusters in the data compliments the other two components in the equation, in particular the correlation coefficient ($\alpha$ in Eq. (6)) since $\alpha$ is highly vulnerable to very distinct clusters of points aligned

on a diagonal axis. This can result in high correlation coefficient values in spite of low correlation inside the individual clusters inevitably misleading the model calibration. The separated clusters often occur in environmental models where different land-use classes and soil classes etc. can produce patchy spatial patterns. The new spatial efficiency metric (optimal value equals to 1) is defined as:

$$SPAEF = 1 - \sqrt{(\alpha - 1)^2 + (\beta - 1)^2 + (\gamma - 1)^2} \tag{6}$$

$$\alpha = \rho(A, B) \text{ and } \beta = \left(\frac{\sigma_A}{\mu_A}\right)\Big/\left(\frac{\sigma_B}{\mu_B}\right) \text{ and } \gamma = \frac{\sum_{j=1}^{n} min(K_j, L_j)}{\sum_{j=1}^{n} K_j}$$

where $\alpha$ is the Pearson correlation coefficient between observed AET map (A) and simulated AET map (B) for a particular

month, $\beta$ is the fraction of coefficient of variations representing spatial variability and $\gamma$ is the percentage of histogram intersection (Swain and Ballard, 1991). The gamma ($\gamma$) is calculated for a given histogram $K$ of the observed AET map (A) and the histogram $L$ of the model simulated AET map (B), each one containing $n$ bins i.e. herein 100 bins. The maps are standardized to a mean of 0 and a standard deviation equal to 1 (zscore) to avoid the effect of different units. In this study, we compare AET from TSEB (in W/m$^2$) based on instantaneous satellite data with daily averaged AET (mm/day) simulated

by the model. It should be noted that we also examined whether we could improve the discriminative skill of the SPAEF using the slope of the QQ plot instead of histogram intersection. However, all attempts and numerous other spatial metrics including Mapcurves, FSS, Goodman and Kruskal's lambda, Theil's Uncertainty, EOF and Cramér's V (Cramér, 1946; Koch et al., 2015; Rees, 2008) did not distinguish the general AET patterns as well as the spatial efficiency (SPAEF) metric. The strength of the spatial efficiency metric is that each component contains different and non-overlapping information.

Moreover, the components are straightforward as compared to the aforementioned metrics. While the correlation term ($\alpha$) expresses only the spatial correlation of AET values, the coefficient of variation term ($\beta$) expresses only the range/contrast in the image while the histogram term ($\gamma$) only expresses the agreement on histogram shape without considering either variation or correlation. Since all three terms are bias-insensitive, the spatial efficiency only constrains the model simulations with the pattern information in the satellite data while leaving the water balance (bias) to be constrained by streamflow





metrics. The readers are referred to our subsequent study by Koch et al. (2017a) where we elaborate and rigorously analyse every component of this metric as well as compare it with other spatial metrics.

**4.2 Sequential screening of the model parameters**

We applied the variance-based sequential screening (SS) method based on Morris elementary effects. This method was
firstly introduced by Cuntz et al. (2015) to identify, with a low computational budget, the parameters which are most informative regarding a certain model output M. The method can be understood as a pre-processor to different kinds of many-query applications such as model calibration, Sobol sensitivity analysis and parameter uncertainty estimation. These methods can then be performed using only the set of informative parameters while the uninformative parameters are discarded and fixed at a default value.

For this approach the parameters are sampled in trajectories as initially described by Morris (1991) and improved by Campolongo et al. (2007). Each trajectory consists of (N+1) parameter sets assuming that N is the total number of model parameters. The first parameter set in each trajectory is sampled randomly while all the subsequent sets *i (i>1)* differ to the prior set (*i-1*) in exactly one parameter value. Therefore, the whole trajectory is a path through the parameter space. Trajectories allow us to sample the whole parameter space efficiently and consider parameter interactions to certain extend.

In the approach of Cuntz et al. (2015), only a small number ($M_1$) of such trajectories are sampled to lower the computational burden. The resulting ($M_1$ x (N+1)) model outputs are derived and the elementary effects (EE) are computed for each parameter. The EE's are then used to identify the most informative parameters by deriving a threshold splitting the parameters into a set $N_u$ of uninformative and a set $N_i$ of informative ones. In the following, the first parameter set is again sampled randomly but then only the uninformative parameters are perturbed meaning that the new trajectory only consist of
($N_u$+1) parameter sets. The derivation of model output and calculation of EE's is repeated. The major step is to determine whether one of the previously uninformative parameters is now above the threshold and if so it is added to the set of informative parameters $N_i$. These steps are repeated until no further parameter is added to the set $N_i$. At the end $M_2$ trajectories are sampled to confirm that the set of uninformative parameters $N_u$ is stable and no further parameter would be found to be informative. Also a combination of Latin-hypercube global sampling strategy and a local sensitivity method (van
Griensven et al., 2006) is tested to further reduce the number of effective parameters for our subsequent study by Koch et al. (2017a).

**4.3 Model calibration and validation**

We calibrated the 1 km-daily mHM for the Skjern basin in Denmark using the well-known global search algorithm Shuffled Complex Evolution University of Arizona (SCE-UA) (Duan et al., 1992). The SCE-UA algorithm is configured with two
complexes running in parallel with 53 (2n+1) parameter sets in each complex and 27 (n+1) parameter sets per sub-complex. Moreover, the maximum relative objective function change is set to 1% over five iterations as the model convergence criterion. This criterion was usually reached after 3500 runs; in rare cases up to 8000 runs were necessary. We evaluated the differences between monthly AET estimates from the TSEB reference data and simulated AET from the hydrologic model for the calibration period (2001-2008) and validation period (2009-2014) using the same cloud-free days in summer.

The two streamflow stations are defined separately to follow the improvements in each metric throughout the calibrations. After testing different combinations of streamflow and spatial metrics, we chose two streamflow metrics (KGE and $KGE_{low}$) and one spatial efficiency metric given by Eq. (5) and (6), respectively. These objective functions are used individually or combined in three model calibration cases based on (1) only streamflow using equally-weighted KGE and $KGE_{low}$, (2) only spatial patterns of AET using spatial efficiency, (3) both equally-weighted streamflow and spatial pattern match using all
three metrics. It should be noted that the case 2 calibration is designed as a benchmark to explore how good the pattern match can get when not considering streamflow performance, even though the solution might not be interesting from a




hydrological perspective, since the bias insensitive spatial pattern metric does not secure a reasonable water balance. To test the overall robustness of the calibration framework we use an ensemble of nine calibrations for case 1 and nine calibrations for case 3 each started from a different seed number. In order to fairly weigh the objective functions, we retrieve the residuals ($\varepsilon$) from the three objective functions based on a random initial model run (Eq. 7, 8 and 9). We calculate the new

weights which will result in equal contribution to the phi$_{total}$ ($\Phi_{total}$) i.e. 50% from spatial metric and 50% from the two streamflow metrics. Ideally if it exists the optimizer searches a parameter set resulting in zero phi$_{total}$ otherwise the closest point to zero will be considered as optimum solution.

$$\Phi_Q = \sum_{i=1}^{2} \left( \varepsilon_{KGE_i} * \omega_{KGE,i} \right)^2 + \sum_{i=1}^{2} \left( \varepsilon_{KGE_{low,i}} * \omega_{KGE_{low,i}} \right)^2 \tag{7}$$

$$\Phi_{Spatial} = \sum_{m=1}^{6} \left( \varepsilon_{SPAEF_m} * \omega_{SPAEF_m} \right)^2 \tag{8}$$

$$\Phi_{total} = \Phi_Q + \Phi_{Spatial} \tag{9}$$

Where $\Phi_Q$ is the total phi for streamflow of the two gauges A and –B and $\Phi_{Spatial}$ is the total phi for spatial performances of six summer months. For Q-only calibration, the weight for SPAEF ($\omega_{SPAEF}$) becomes zero whereas for spatial-only

calibration the weights for KGE and KGE$_{low}$ become zero.

## 5 Results

### 5.1 Sequential screening of the model parameters

Table 2 shows the sequential screening results based on KGE, KGE$_{low}$ and SPAEF, respectively. Each objective function reflects on different spatio-temporal dynamics of the catchment. While KGE and KGE$_{low}$ evaluate high and low streamflow

dynamics and biases, the bias-insensitive SPAEF focuses on only spatial patterns of AET. From the results it is clear that some of the highly sensitive parameters for streamflow dynamics especially interflow-related parameters, groundwater-related geology parameters and single routing parameter have minor to zero influence on the spatial patterns of AET. The new ET parameters, ET$_{ref}$–ap (pervious), –af (forest), –b and –c are identified to be informative based on all objective functions. The root fraction coefficient for forest (rotfrcoffore) appeared to be not very important for streamflow metrics

whereas it is crucial for SPAEF. Similarly the two newly introduced parameters i.e. root fraction coefficient for sand and clay (i.e. rotfrcofsand and rotfrcofclay) soil are informative based on all three objective functions. Organic matter for forest (orgmatforest) is especially important for low flows whereas and organic matter for impervious areas (orgmatimper) has zero influence on spatial patterns of AET. The exponent slow interflow (expslwintflw) parameter is found to be most informative for low flows while recharge coefficient (rechargcoef) is most informative for streamflow and ET$_{ref}$–af is most informative

for calibrating spatial patterns of AET.

On average 475 model evaluations are required to split the total number of 48 parameters into informative and uninformative ones. However, the number of iterations is dependent on objective function, therefore; 449 model runs were required for KGE, 431 model runs for KGE$_{low}$ and 544 model runs for SPAEF. This is in close agreement with the computational budget of 10N model evaluations already reported by Cuntz et al. (2015). This also makes the sequential screening method

computationally very attractive compared to other global search methods. However, the computational advantage is at the cost of exploring a larger part of the parameter space, hence the sequential screening is mostly valuable for identifying informative/non-informative parameters prior to calibration or further assessment of the parameter behaviour. Overall, these results show that there are 26 parameters above the threshold of 1% of at least one case (Table 2). In principal the parameters with zero sensitivity (SPAEF column) can be fixed at some value during calibration which may lead to faster convergence





with lower degree of freedom. However, we include the same set of 26 parameters in all three calibration cases for consistency.

**Table 2 Selected 26 parameters for calibration and their normalized sensitivity indices sorted based on SPAEF column. Zero values are highlighted with a grey shade.**

| Parameter | Description | Normalized Sensitivity | | |
| --- | --- | --- | --- | --- |
| | | KGE | KGElow | SPAEF |
| ETref-af | Intercept – forest | 0.022 | 0.117 | **0.646** |
| ETref-c | Exponent coefficient | 0.031 | 0.732 | 0.490 |
| ETref-b | Base coefficient | 0.439 | 3.013 | 0.317 |
| rotfrcoffore | Root fraction for forest areas | 0.011 | 0.013 | 0.162 |
| ETref-a | Intercept – nonforest (pervious) | 0.308 | 3.235 | 0.157 |
| ptfhigconst | Constant in Pedo-transfer function for soils with sand content higher than 66.5% | 0.063 | 0.223 | 0.096 |
| rotfrcofclay | Threshold for actual ET reduction for clay | 0.101 | 0.274 | 0.094 |
| ptfhigdb | Coefficient for bulk density in Pedo-transfer function for soils with sand content higher than 66.5% | 0.036 | 0.257 | 0.070 |
| rotfrcofsand | Threshold for actual ET reduction for sand | 0.120 | 0.439 | 0.061 |
| canintfact | Canopy interception factor | 0.004 | 0.029 | 0.018 |
| orgmatforest | Organic matter content for forest | 0.136 | 0.893 | 0.014 |
| ptfhigclay | Coefficient for clay content in pedo-transfer function | 0.008 | 0.033 | 0.011 |
| infshapef | Infiltration shape factor | 0.103 | 0.099 | 0.006 |
| ptfkssand | Coefficient for sand content in pedo-transfer function for hydraulic conductivity | 0.415 | 2.780 | 0.002 |
| ptfksconst | Constant in pedo-transfer function for hydraulic conductivity of soils with sand content higher than 66.5% | 0.236 | 0.842 | 0.001 |
| snotrestemp | Snow temperature threshold for rain and snow separation | 0.034 | 0.206 | 0.000 |
| ptfksclay | Coefficient for clay content in pedo-transfer function for hydraulic conductivity | 0.040 | 0.313 | 0.000 |
| orgmatimper | Organic matter content for impervious zone | 0.009 | 0.020 | 0.000 |
| expslwintflw | Exponent slow interflow | 0.412 | **3.490** | 0.000 |
| slwintreceks | Slow interception | 0.872 | 1.296 | 0.000 |
| intrecesslp | Interflow recession slope | 0.602 | 1.105 | 0.000 |
| rechargcoef | Recharge coefficient | **0.935** | 0.666 | 0.000 |
| geoparam1 | Parameter for geological formation 1 | 0.328 | 0.138 | 0.000 |
| geoparam2 | Parameter for geological formation 2 | 0.558 | 0.207 | 0.000 |
| strcelerity | Streamflow celerity for routing | 0.364 | 0.062 | 0.000 |
| intstorcapf | Interflow storage capacity factor | 0.198 | 0.010 | 0.000 |

**5.2 Model calibration and validation**

The mHM model is calibrated using streamflow records (gauges A and B in Figure 1) from eight years (2001-2008) and validated for a recent period (2009-2014). Preceding four-years of these two periods (1997-2000 and 2005-2008) are used for model warm-up. We prepared remotely sensed monthly averaged AET pattern-maps calculated for these years



considering only cloud-free days from summer months. AET patterns of winter months are not considered since it is mostly cloudy and ET is very low and uniform (energy limited) in winter.

The 26 selected parameters from SS are used in the following three calibration strategies: 1) only streamflow oriented (Q-only) calibration using equally-weighted KGE and $KGE_{low}$, 2) only spatial pattern oriented calibration using SPAEF and

3) streamflow and spatial patterns of AET together using all three objective functions with equal weights of 50% on spatial metric and 50% for the two streamflow metrics (25% each). Table 3 provides the overall picture of the three different calibration strategies where two of these strategies are based on an ensemble of nine calibrations. Therefore, the basic descriptive statistics are also given as robustness indicators. The results show that the combined calibration (Q and Spatial) produces similar results to both Q-only and Spatial-only calibrations focusing on streamflow and spatial patterns of AET

respectively; whereas the single metric calibrations gave very different results for the opposite objective functions e.g. SPAEF vs streamflow metrics. It is interesting that when comparing the calibration ensemble with the median performance there is very limited trade-off between the Q-only and the combined Q and Spatial calibrations which have very similar average KGE values. When looking specifically at the best performing ensemble member (lowest total phi), there is a more pronounced trade-off between the Q-only and Q and Spatial together calibrations, as the streamflow performance is poorer

when SPAEF is included in the group of objective functions. The differences in the streamflow metrics indicate that each objective function carries relevant but slightly conflicting information. Moreover, the results show that the hydrologic model simulates best AET patterns in different months for different ensemble calibrations. In other words, while one ensemble member has the best performance for April, other calibrations may have the best performance for May and June. This is a secondary trade-off which illustrates that the calibration might benefit from temporal variability in the parameters controlling

the spatial parametrization scheme. It should be noted that ranking the calibrations within the two ensembles are based on the overall phi that is comprised of all objective functions for the corresponding calibration. For that reason, the best member of Q and Spatial calibration holds the lowest (i.e. best) phi of 0.91 comprised of the highest possible KGE, $KGE_{low}$ and SPAEF at the same time but not necessarily the highest SPAEF alone. This resulted in a slightly lower SPAEF mean of 0.395 for the best member compared to the median member with a SPAEF mean of 0.396 (Table 3).

**Table 3 Summary of the calibration results for three cases. Median and standard deviation (Std.) refers to the calibration ensemble ranked based on their total phi. Lower phi indicates lower error.**

| Metrics | | Q-Only | | Spatial-Only | Q and Spatial | |
|---|---|---|---|---|---|---|
| | Gauge | Median (Std) | Best | Single Cal. | Median (Std) | Best |
| KGE [-] | (A) | 0.83 (0.03) | 0.97 | -1.47 | 0.88 (0.01) | 0.89 |
| KGE [-] | (B) | 0.94 (0.02) | 0.97 | -1.03 | 0.92 (0.01) | 0.93 |
| $KGE_{low}$ [-] | (A) | 0.81 (0.02) | 0.86 | -3.12 | 0.81 (0.02) | 0.82 |
| $KGE_{low}$ [-] | (B) | 0.81 (0.02) | 0.85 | -2.66 | 0.79 (0.02) | 0.8 |
| BIAS [%] | (A) | -6.24 (2.23) | -0.98 | 38.56 | -2.44 (0.62) | -1.25 |
| BIAS [%] | (B) | 1.51 (0.83) | 1.83 | 46.87 | 1.42 (0.93) | 3.86 |
| Apr - SPAEF | | -0.88 (0.33) | -0.17 | 0.5 | 0.57 (0.02) | 0.51 |
| May - SPAEF | | -0.62 (0.26) | -0.12 | 0.47 | 0.35 (0.07) | 0.35 |
| Jun - SPAEF | | -0.59 (0.23) | -0.09 | 0.36 | 0.27 (0.11) | 0.27 |
| Jul - SPAEF | | -0.39 (0.10) | -0.36 | 0.51 | 0.38 (0.04) | 0.43 |
| Aug - SPAEF | | -0.29 (0.10) | -0.36 | 0.53 | 0.48 (0.05) | 0.49 |
| Sep - SPAEF | | 0.02 (0.19) | 0.30 | 0.40 | 0.33 (0.02) | 0.32 |
| $Phi_{total}$ | | 0.24 (0.06) | 0.11 | 0.52 | 0.94 (0.03) | 0.91 |

The results of the Q-only model calibration using only KGE and $KGE_{low}$ reveal very poor simulated patterns of AET, with negative SPAEF for all months. This is not surprising since that optimization is not constrained regarding the spatial





patterns, but also illustrates that discharge observations alone contain no spatial pattern information of AET. In contrast, the spatial-only calibration using only SPAEF shows a very poor water balance, with negative KGE and large bias. We are aware that spatial-only calibration is not applicable and meaningful for hydrologic studies.

The model performance development through the calibrations (9+9) and optimum points are shown using scatter plots in
5    Figure 2, which displays all model runs with phi values inside the specified plot ranges. The scatter plots illustrate trade-offs between objective functions and consistency among calibration ensemble members. The performance regarding spatial patterns (phi$_{Spatial}$) displays a high degree of trade-off with all combined calibrations achieving phi$_{Spatial}$ values around 0.8 whereas the Q-only calibrations achieve phi$_{Spatial}$ values ranging 2.8 and 4.4.There are two main clusters in the Q-only calibrations one around 0.11 phi$_Q$ and the other around 0.25 phi$_Q$ whereas all nine Q and spatial calibrations follow a similar
10   level in y axis (phi$_{Spatial}$). It is surprising to see that SCE-UA did not always find the same optimum solution with varying seed number, which is the case mainly for the Q-only calibration. This raises the question whether SCE-UA algorithm may have shortcomings e.g. a local minimum since we applied the same stopping rules to all calibrations but selected different seed numbers only. Perhaps more consistent optimum solutions for the Q-only calibrations could have been achieved with tighter stopping rules and the same initial parameter sets.

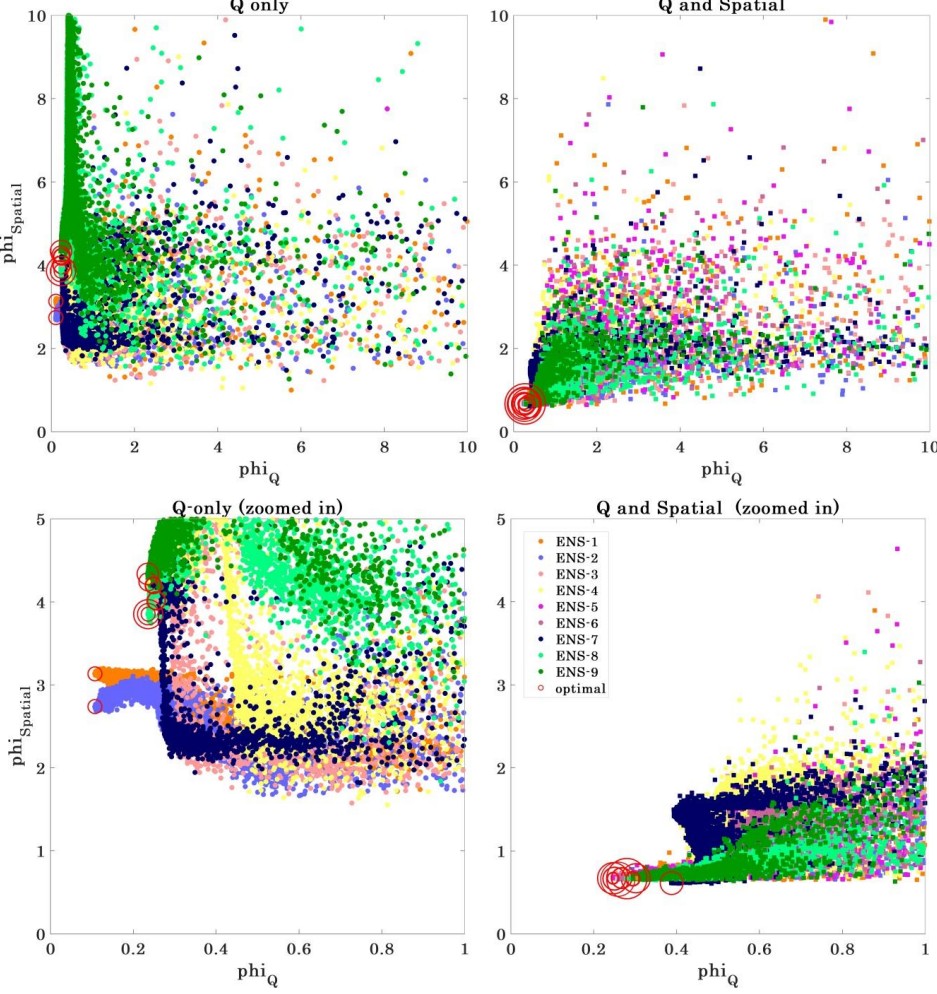



**Figure 2. Scatter plots of total spatial phi versus total Q phi for all nine calibration ensemble members. First and second row subplots are the same figures except for different extent i.e. [10 10] and [1 5] to zoom into the edge of the search space. Different radius of red circles is used to show the optimum points for all nine ensemble members clearly.**

5    Similarly, the grey shades in Figure 3 show the ensemble range of simulated hydrographs for the Q-only and Q and Spatial calibrations. From the hydrographs it is clear that the ensemble range for station A is generally larger than station B, indicating larger uncertainty for sub-basin A. Interestingly, Figure 3 also illustrates that the Q and Spatial calibration reduces the uncertainties, not only in AET simulations, but also in streamflow simulations, as indicated by the slightly narrower range in simulated streamflow for the Q and Spatial calibrations. However, even though the range of hydrographs is slightly

10   narrower the simulations are also further from the observed during summer months.

**Figure 3. Average hydrograph of all years in the calibration period (2001-2008) to illustrate the ensemble of nine model**
15   **calibrations with different seed numbers.**



The corresponding simulated AET maps for the results presented in Table 3 are shown in Figure 4. Only the best performing members from the two ensembles are presented in this figure. The maps are normalized with their mean value to use one representative colour bar in the legend. As indicated in Table 3, the resultant maps from Spatial-only (third row in Figure 4) and Q and Spatial calibrations (fourth row in Figure 4) are obviously more similar to the reference monthly maps (first row

in Figure 4) than the maps of Q-only calibration (second row in Figure 4). The results clearly show that the model can simulate month-to-month variations in AET patterns reasonably well. The poor AET performance in the Q-only maps is obvious in the second row of Figure 4 where we see only a uniform simulated AET pattern except for the forest areas revealing very little information about variability in AET and the influence of soil and vegetation. This is from the fact that the KGE and $KGE_{low}$ objective functions contain no information on the patterns of AET resulting in an unconstrained

optimization regarding spatial pattern and variability. Therefore the optimizer randomly moves in the SPAEF solution space and picks the best streamflow performance with no regard to AET patterns. Although not perfect (average SPAEF = 0.46 and 0.40), the simulated pattern match in the last two rows of Figure 4 is quite good compared to the remote sensing based estimate since the simulation is able to represent the general pattern influenced by soil, vegetation and land cover while maintaining a similar variance and smoothness.

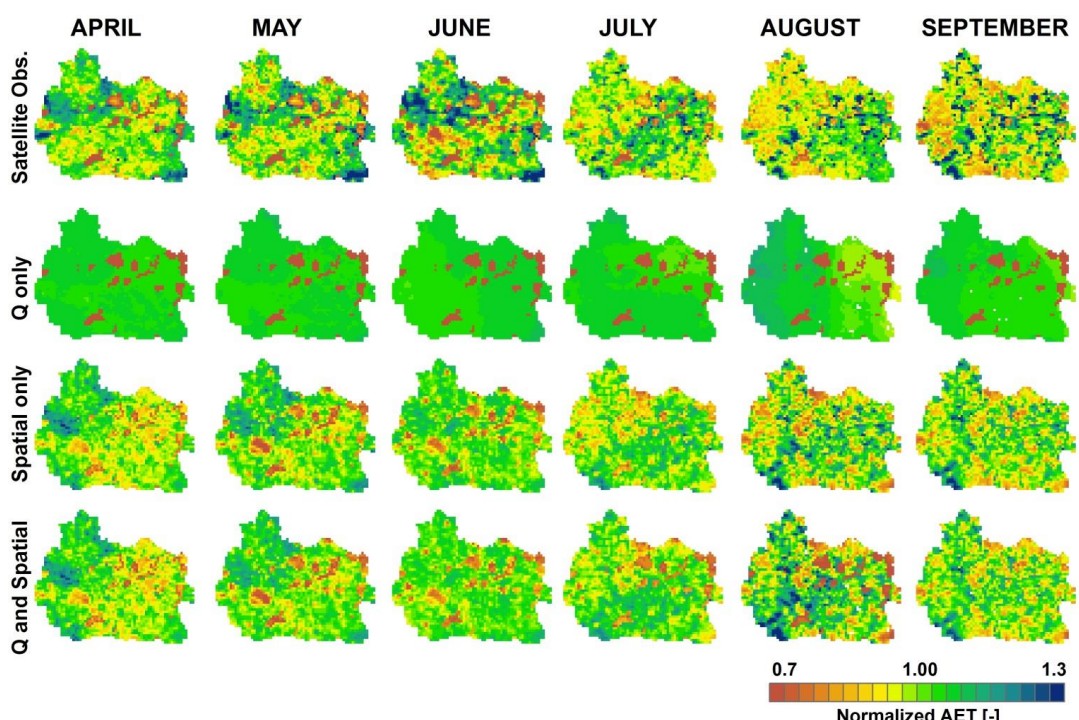

**Figure 4. Three different calibration strategies: 1) streamflow-only (second row) 2) spatial-only (third row) and 3) streamflow and spatial together (fourth row) are compared with monthly TSEB estimates (first row). Calibrations are evaluated for monthly averages from April to September using cloud-free days. Note that these maps are normalized with their mean to use one**
**representative colour bar and highlight the pattern information.**

Table 4 shows the same results as Table 3 but for the validation period spanning from 2009 until 2014. Obviously, the results are somewhat poorer than those for the calibration period. A prop in performance for spatial-only and combined metrics is mainly seen for KGElow and the total bias, whereas the SPAEF for Spatial-only and Q and Spatial remain similar to the
calibration periods with average SPAEF around 0.4. Interestingly, there is no real trade-off for streamflow metrics between



Q-only and Q and Spatial calibrations for the validation period, even for the best performing ensemble member. Although a better streamflow performance could be achieved by Q-only calibration during calibration, this cannot be sustained during validation, indicating some overfitting when using streamflow metric only. In contrast, the SPAEF performance does not drop during validation for the combined Q and Spatial optimization, indicating less overfitting and a more robust model
parametrization.

**Table 4 Summary of the validation results for three cases. Median and standard deviation (Std.) refers to the validation ensemble ranked based on their total phi. Lower phi indicates lower error.**

| Metrics | | Q-Only | | Spatial-Only | Q and Spatial | |
|---|---|---|---|---|---|---|
| | Gauge | Median (Std) | Best | Single Cal. | Median (Std) | Best |
| KGE [-] | (A) | 0.83 (0.01) | 0.86 | -1.65 | 0.86 (0.01) | 0.88 |
| KGE [-] | (B) | 0.89 (0.02) | 0.93 | -1.40 | 0.87 (0.01) | 0.88 |
| $KGE_{low}$ [-] | (A) | 0.70 (0.04) | 0.79 | -3.84 | 0.72 (0.02) | 0.76 |
| $KGE_{low}$ [-] | (B) | 0.65 (0.02) | 0.66 | -3.72 | 0.64 (0.03) | 0.65 |
| BIAS [%] | (A) | -15.76 (2.26) | -10.38 | 29.64 | -11.89 (1.15) | -8.85 |
| BIAS [%] | (B) | 5.23 (1.04) | 5.39 | 55.86 | 5.00 (1.45) | 9.13 |
| Apr - SPAEF | | -0.76 (0.30) | -0.11 | 0.47 | 0.51 (0.02) | 0.53 |
| May - SPAEF | | -0.65 (0.28) | -0.09 | 0.56 | 0.51 (0.03) | 0.51 |
| Jun - SPAEF | | -0.50 (0.19) | -0.13 | 0.38 | 0.27 (0.04) | 0.27 |
| Jul - SPAEF | | -0.56 (0.17) | -0.30 | 0.59 | 0.48 (0.09) | 0.50 |
| Aug - SPAEF | | -0.15 (0.10) | -0.33 | 0.18 | 0.19 (0.07) | 0.21 |
| Sep - SPAEF | | -0.12 (0.13) | -0.31 | 0.44 | 0.35 (0.02) | 0.37 |
| $Phi_{total}$ | | 0.73 (0.11) | 0.50 | 0.60 | 1.38 (0.10) | 1.34 |

## 6 Discussion

In the initial phase of the study numerous flawed calibrations were carried out in an attempt to produce simulated spatial
patterns of AET similar to the satellite based reference patterns. However, the inability to produce similar patterns was found to be caused by limitations in spatial model parameterisation and spatial performance metric choice. Regarding the spatial parameterisation, the initial model was based on a spatially uniform parameterisation of root fraction coefficient and PET correction factor, two parameters with major control on the simulated AET. Therefore, more flexible yet physically meaningful parameterisations were implemented where full spatial variability was enabled by combining 2-3 calibration
parameters to initial spatial distributions of soil type and LAI. Regarding the use of appropriate spatial performance metrics, the initial attempts using standard metrics of correlation coefficient, Mapcurves (Hargrove et al., 2006), coefficient of variation, Goodman and Kruskal's lambda (Goodman and Kruskal, 1954), agreement coefficient (Ji and Gallo, 2006), Theil's Uncertainty, EOF and Cramér's V (Cramér, 1946; Koch et al., 2015; Rees, 2008) proved to be inadequate in a calibration framework, since undesired visual patterns were achieved e.g. with high correlation, but too low standard
deviation or highly separate clusters. Therefore, we developed the SPAEF metric which proved to be very efficient for calibrating the model to a satisfying spatial pattern by combining correlation coefficient, coefficient of variation ratio and histogram overlap in a robust metric that guides the model calibration well. For more details of the testing of the SPAEF in a calibration framework please see the study by Koch et al. (2017a). It is our experience and recommendation that incorporating the spatial dimension in all aspects of the distributed hydrological model development from model structure,
parametrization, metric selection, sensitivity analysis and calibration is essential in order to achieve significant improvement in the spatial pattern performance of a model. It is recognized that traditional downstream discharge measurements contain much more accurate and robust information on the overall water balance compared to the non-continuous remotely sensed estimates, and therefore, the model constraint on biases should only originate from these streamflow observations.





Conversely, it is well-known that aggregated streamflow measurements contain no information on spatial patterns upstream of the measurement (Stisen et al., 2011b). Therefore, the combination of satellite derived patterns and aggregated streamflow measurements are an ideal way of constraining distributed hydrological models. In fact spatial patterns should always be considered when evaluating distributed models. Even if detailed satellite estimates are not available, expert judgments and

land cover information should be used to select the most appropriate parameter set (producing the most likely spatial patterns) among equally likely solutions obtained through discharge-only calibration. When a distributed model is applied, it should be desired, that it produces not only satisfying discharge simulations, but at the same time produces realistic spatial patterns of states and fluxes such as AET and soil moisture. White et al. (2017) also highlighted the importance of getting the spatial patterns right in their study since constraining the model against streamflow alone did not secure a robust land cover

change scenario modelling.

The monthly spatial maps are built based on the AET patterns from cloud-free days. Here, we ignore the temporal aspect and focus only on the consistent spatial patterns for each month of the growing season. The advantage of this approach is that only the main information content of the satellite data, their spatial patterns, are utilized while the uncertainty associated to the absolute values of the AET estimates are not influencing the calibrations. In addition, the simulated monthly mean AET

maps reflect mainly the model parameterisation and to a lesser degree the day-to-day variation in climate forcing. This is desirable since the aim of the model calibration is to optimize the model parameterisation with a given climate forcing dataset. The current calibration framework builds on the assumption that the satellite based estimate of AET patterns approximate an observed patterns that is suitable for model optimization.

The calibration results obtained in the current study where three strategies were tested with varying combinations of

objective functions showed that with an appropriate metric design, limited trade-offs can be achieved when combining streamflow and spatial pattern metrics in a joint calibration framework. This is largely attributed to the fact that by design, the spatial performance metric is bias-insensitive whereas the streamflow metrics have very little sensitivity to spatial redistribution of AET patterns as long as the spatial averages remain unchanged. Bias and temporal variability of satellite derived AET estimates could also be useful for model optimization, however, in this study, we deliberately limited the

information content of the satellite data to address the spatial patterns. This is done because even though the satellite based AET estimate is validated against eddy covariance stations (Mendiguren et al., 2017) they only represent specific cloud-free days limiting their value to assess the long term water balance of the catchment. The calibration results using only streamflow metrics revealed that this traditional calibration target cannot guarantee satisfying spatial pattern performance even though the model structure and parametrization framework enables this without much compromise as illustrated by the

performance of the combined Q and Spatial calibrations which resulted in very similar performance of both streamflow and spatial patterns as the single objective calibrations individually.

The spatial model parameterisation applied in Skjern catchment can be domain-specific due to its very sandy soils whereas the calibration framework and SPAEF metric can be applied to any other river basin in the world. Although we found spatial parametrization examples from different counties such as Sweden (Wiklert, 1961) and Germany (Vetter and Scharafat,

1964), we have not tested our field capacity dependent root fraction approach under other conditions yet. This will be a topic of our subsequent study on multiple European catchments. Regarding the dynamic scaling function, developed for incorporating remotely sensed LAI in $ET_{ref}$ scaling, it should be noted that the use of LAI to describe the deviation of each grid cell from the assumed reference grass is a simplification. Albedo could also have been included in the dynamic scaling function; however, one could argue that albedo and LAI are somewhat correlated and including one of them is already

contributing the information about the other (Chen et al., 2005; Liu et al., 2017; Stisen et al., 2008). Moreover, we limit this study to temporally averaged spatial patterns of AET and deliberately choose to ignore the day-to-day dynamics of AET. In this study, spatially varying but temporally constant field capacity dependent root fraction is utilized, however; it would be more elegant and physically more sound to represent the seasonality in root-growth dynamics more realistically by





implementing a seasonally varying root fraction coefficient (beta) that is similar to the concept of LAI based PET correction using the DSF module.

## 7 Conclusions

Our study aimed at parameterising a distributed hydrologic model for simulating distributed actual evapotranspiration patterns before an ensemble calibration using satellite based data. This order is crucial for a progressive hydrologic modelling with flexible model structure based on open-source philosophy. All these steps should be suitable for the catchment to give model enough flexibility to adjust to pattern observations. The calibration efforts will have limited effect on spatial patterns if the model parameterisation has not been investigated with pattern performance in mind. Ideally, the models should offer different parametrization schemes or at least have room for development based on open-source philosophy so that we can test different spatial parameterisations for a particular calibration goal. Here, we implemented a field capacity dependent root fraction coefficient determining the root profile over depth for different soil and vegetation types. We introduced a dynamic scaling function which imprints the leaf area index in the potential evapotranspiration. After organizing the spatial parameterisation of the model in a parsimonious manner, we also reduced the number of parameters using sequential screening. Only the informative parameters from the sequential screening are used in the subsequent ensemble calibration exercise. We then assessed the effect of different calibration strategies including monthly spatial patterns of actual evapotranspiration in combination with traditional streamflow observations. In the spatial calibration, the agreement between observed and simulated spatial patterns is added as a part of the objective function used for model optimization. For that a multi-component bias in-sensitive spatial efficiency metric (SPAEF) is used to evaluate the simulated AET maps. The following conclusions can be drawn from our results:

- Preparing the model parameterisation for spatial calibration is a key element for achieving the calibration objectives. More specifically, the model parameterisation needs to be designed to allow the spatial parameter distribution to be optimized through calibration.

- The newly proposed spatial efficiency metric (SPAEF) is proven to be robust and easy to interpret due to its three distinct and complementary components of correlation, variance and histogram matching.

- Based on the multi-component calibration results, including spatial pattern information in calibration, significantly improves the spatial model simulations while maintaining similar streamflow performance. For the combined calibration, there is a limited trade-off between streamflow and spatial patterns for the best performing calibration ensemble compared to the Q-only calibration. However, this trade-off disappears in the validation test, indicating that a more robust parameter set is achieved during the combined Q and Spatial calibration.

Overall, the hydrological modelling community can benefit from building familiarity with several aspects of spatial model evaluation, including spatial parameterisation and multi-component spatial performance metrics.

## Acknowledgements

We acknowledge the financial support for the SPACE project by the Villum Foundation (http://villumfonden.dk/ ) through their Young Investigator Programme (grant VKR023443). The TSEB code is retrieved from https://github.com/hectornieto/pyTSEB. All MODIS data was retrieved from the online Data Pool, courtesy of the NASA Land Processes Distributed Active Archive Center (LP DAAC), USGS/Earth Resources Observation and Science (EROS) Center, Sioux Falls, South Dakota, https://lpdaac.usgs.gov/data_access/data_pool. Pre-processing ET with crop coefficient type dynamic scaling function is available in the mHM version 5.7 (www.ufz.de/mhm/). The Python and Matlab scripts for SPatial EFficiency (SPAEF) and a tutorial are available in the SPACE project website (http://www.space.geus.dk/).





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
