# Peer review of "Combining satellite data and appropriate objective functions for improved spatial pattern performance of a distributed hydrologic model"

_Hydrology and Earth System Sciences, 2017_

## Referee Comment (RC1) · H. Bogena (Referee) · 20 Oct 2017

This paper deals with an innovative calibration framework that combines temporally aggregated observed spatial patterns with a new spatial performance metric and a flexible spatial parameterisation scheme. An application of the mesoscale Hydrologic Model (mHM) to the Skjern River Basin is used as an example show the effectiveness of the presented calibration framework. This a very timely topic that fits very well to the scope of HESS.

However, the spatial model parameterizing methodology is not well described. Both the root fraction coefficient and the PET correction factor parameterizations should be

presented in more detail including graphical presentations of the underlying relationships.

For these reasons, my recommendation is to accept this manuscript with minor revisions. I have provided specific comments and suggestions for improvement below.

Specific comments:

P3L7: Which models?

P3L16-32: This section is redundant with the method section and should be shortened.

P5L6 Delete "s"

P5L9-11: How accurate are the AET maps, e.g. in relation to continuous EC-measurements?

P5L30-31: Why should this procedure accelerate model runs?

P6L8: "stretch the spatial contrast" of what?

P6L8: "...based on soil and vegetation properties..."

P6L11: Please change "domain-specific" into "site-specific" or "local"

P6L11-12: According Feddes et al. (2001) and others, the root fraction coefficient is a vegetation dependent coefficient. Please add more information on why your assumption that this coefficient can be explained by field capacity is justified.

P6L14-15: At this point it unclear why these parameterizations increase the model freedom. Please reorganise the text in a more comprehensively way.

P6L19-24: It took me some time to understand your method, also because the assumption that the root fraction is linked to FC is counterintuitive. For a better understanding of your method, it would be helpful to graphically show the relationship between FC and rooting depth based on the soil database.
P6L22: Why do you restrict your method to pervious non-forested areas?

P6L22: Please change "domain-specific" into "site-specific" or "local"

P7L2: Why should soil properties of sandy soils impede root development?

P7L2-3: Which parameters and why is a fine vertical discretization more effective?

P7L5-7: By changing the root fraction parameters for maximum and minimum FC, the relationship between FC and rooting depth will be changed. In the extreme case, both parameters have the same value, which means that there is no relationship at all. Did you check whether the optimised model still provides realistic relationships between FC and rooting depth? In addition, it is unclear, how you derived FC values from your digital soil map. Typically, one would sum up the horizontal FC values down to the certain soil depth (e.g. rooting depth). Please provide additional information.

P7L16-17: You should not mention equations that have not been already introduced.

P7L21-25: The section needs to be rewritten in a more comprehensible way.

P7L26: It is how this equation was derived and why it should be "physically meaningful". In addition, it is unclear how the DSF parameter is used to correct ET_ref.

P7L28: The previous sections also belong to "Methods".

P9L1: Only accepted paper should be used a reference.

P9L4-9: This section can be omitted.

P9L24-26: Is this statement relevant for this work or Koch et al., 2017a

P9L34: Why are you using the "same" cloud-free days?

P10L5: Either use the Greek symbol for phi or "phi" (here and elsewhere)

P10L22: Delete "and"

P10L33-34: I wonder why the parameter "root fraction" for forest area is listed and not

for the impervious non-forest areas, since the latter was used for the spatial parameterization.

P10L34: "(SPAEF column in Tab. 2)"

P11L9-10: Why are you using a second model warm-up period (2005-2008)? This is also not mentioned in the text.

P12L18-20: However, it should be noted that the uncertainty of the spatial AET information from remote sensing is typically larger than the runoff measurements.

P12L28: Delete "that"

P13L2-3: This indicates that the spatial AET information either has large uncertainties or that it has very imitated information on the subsurface properties due to extensive irrigation in the catchment. Please discuss.

P15L13: Does Figure 4 present a certain year or monthly averages of several years?

P15L14: The differences seem to vary also from month to month. You should quantify the differences in AET patters, e.g. by presenting the variances or variograms.

P17L21: Awkward sentence. Please reformulate.

P17L25: "This was..."

P617L31-32: Unclear why the sandy soil texture should restrict the model parametrisation method.

P617L32: Please change "domain-specific" into "site-specific" or "local"

P18L23: "has proven"

Figures and Tables:

Figure 1: The RS-AET scale should be reversed (red should indicate high AET values)

Table 1: Columns 3, 4 and 5 should be removed as they provide only limited information, which is already presented in the text.

Additional literature

Feddes, R.A., H. Hoff, M. Bruen, T. Dawson, P. de Rosnay, P. Dirmeyer, R.B. Jackson, P. Kabat, A. Kleidon, A. Lilly, and A.J. Pitman, 2001: Modeling Root Water Uptake in Hydrological and Climate Models. Bull. Amer. Meteor. Soc., 82, 2797–2809.

---

## Referee Comment (RC2) · R. Romanowicz (Referee) · 27 Oct 2017

The authors present a methodology aimed to improve predictions of a distributed hydrological model both in time and space. In order to identify model parameters responsible for the spatial predictions, an additional complex objective function is introduced taking into account the match between observed (satellite based) spatial evapotranspiration patterns and those predicted by the model.

The paper is interesting and the approach taken opens a new direction of research towards the use of spatial remote sensed observations to improve predictions of a distributed hydrological model. In particular, the authors applied site-specific parameterizations to increase the flexibility of the description of actual evapotranspiration characteristics (root zone and potential evapotranspiration corrections).

The authors formulated the problem in a deterministic framework, which might be a good introduction to a new approach, but the discussion on uncertainty is missing.

The paper is well written and requires only some clarification of the presented material and a wider discussion of the assumptions taken. I recommend to publish the paper after minor corrections.

Specific comments:

Page 1, line 18: . . . In addition two new site-specific spatial parameter distribution options have been introduced . . .

Page 2, line 19: This is from the fact . . . grammar should be corrected e.g. 'this is because..'

Page 3, Line 17: it is not clear what the authors mean by 'domain'

Page 3, line 22: is it: . . . for comparing spatial patterns of two continuous variables. . .?

Page 5, lines 8-11: what is the uncertainty of AET estimates?

Page 5, line 30: Could you please give more detail about the way monthly AET maps are applied in the model and the disaggregation method used?

Page 6, line 9: The parameterisation introduced is a very interesting way forward and requires a separate paper backed up with field experiments. Could you please give the possible disadvantages of the parameterisation? Even though the parameterisation decreases the number of parameters of a distributed hydrological model, the parameters require a sufficient amount of observations to be properly identified. The question is how to test the parameterisation using very limited and uncertain information obtained from the indirect and fragmented satellite observations. The other question is, how to estimate the uncertainty related to that parameterisation. Some comments would be

welcome.

Page 6, lines 10-24: How sensitive is water storage variability to this parameterisation?

Page 7, lines 2-10: was the model tested on observations and what assumptions must be fulfilled?

Page 7, line 24: was this parameterisation tested on observations? What assumptions are imposed?

Page 8, line 7: Since comparison ... should start from a new line and a new objective function responsible for reproducing spatial patterns should be introduced.

Page 8, lines 10-11: histograms of what?

Page 8, line 24: The AET from TSEB have been treated as error free data – a comment is needed on the possible errors involved.

Page 9, line 31: That criterion might be very misleading when the response surface is flat. The optimisation algorithm might stop in any part of the optimisation range or, most likely at the edge of the parameter range. The authors are asked for a comment.

Page 10, line 8: ...A and B and .... (- B is confusing)

Page 13, line 12: The existence of local minima depends on the form of the objective function which defines the parameter response surface. In the case of a model with 26 parameters the objective function will show local minima. Following the equifinality hypothesis, there are many parameter sets which give the same value of the objective function and therefore it is not surprising that many different optimum solutions can be found.

Page 14, line 8-9: Spatial calibration constraints the solution rather than reduces its uncertainty – the uncertainty was not evaluated.

Page 15, lines 23-24: Could it be explained why the improvement occurs?

Page 16, Discussion: As I understood, the authors did calibration using different sequences of cases. Might it be advantageous if the optimisations Q-only and Spatial-only were applied iteratively?

Page 17, line 13: . . . associated with . . ..

Page 17, line 32: . . . site-specific due to . . ..

Page 17, line 34: . . .different countries . . ..

Page 18, line 28: The conclusion on achieving a more robust parameter set because the trade-off disappears is not well founded and too general after only one validation exercise.

---

## Referee Comment (RC3) · G. Bertoldi (Referee) · 6 Dec 2017

**Review of the Article**

**Combining satellite data and appropriate objective functions for improved spatial pattern performance of a distributed hydrologic model**

**by Mehmet C. Demirel, Juliane Mai, Gorka Mendiguren, Julian Koch1, Luis Samaniego, Simon Stisen**

The paper addresses the relevant topic of using spatial data for the calibration of distributed hydrological models.

The paper explores the tradeoffs between streamflow-based and spatial-based calibrations, illustrating the benefits of combining separate observation types and objective functions. Given the increasing availability of spatial data, to understand the value of a spatial calibration versus a more traditional calibration is a timing and relevant research topic.

Spatial pattern of actual evapotranspiration (AET), obtained with the mesoscale Hydrologic Model (mHM), are calibrated against spatial patterns of AET estimated trough remote sensing (TSEB model).

The major novel point of the paper is to present novel bias-insensitive spatial pattern metric, which exploits the key information contained in the observed patterns, useful for spatially distributed models optimization.

I found the paper valuable and worth to be published. However, I believe some methodological aspects should be further clarified, as detailed in the specific comments.

I suggest a minor revision for the paper. Please revise also carefully English language.

**General comments:**

1. In the paper formulates the hypothesis that "*The current calibration framework builds on the assumption that the satellite based estimate of AET patterns approximate an observed pattern that is suitable for model optimization*".
   On my advice, this hypothesis should be better motivated and the implications better discussed.

   At the end, six monthly mean "climatological" AET maps have been produced. But spatial patterns are time consistent? If the spatial structure of AET changes a lot from time to time, or there is a great inter-annual variability, then it makes little sense to calibrate the model to such patterns. A preliminary analysis on the temporal variability of the observed AET patterns should be done. Only if spatial AET patterns are quite constant over the different years, the proposed calibration approach could be used.

   Moreover, the fact that an AET monthly "climatology" has been used, could be one reason of the limited tradeoff found with the discharge. Could you discuss about this? In this way, you cannot for example identify from AET observation a drought period, which can have an effect

on discharge.

I see later that this choice is explicit in the paper (see 4.1 Objective functions). "Since all three terms are bias-insensitive, the spatial efficiency only constrains the model simulations with the pattern information in the satellite data while leaving the water balance (bias) to be constrained by streamflow".

However, this implies a hypothesis of spatial invariance of AET patterns. You need to support and discuss this assumption in the paper.

2. Another assumption is that "*Bias and temporal variability of satellite derived AET estimates could be useful for model optimization, however, in this study, we deliberately limited the information content of the satellite data to address the spatial patterns.*"
You clearly separate in your calibration framework the source of temporal information (discharge) from the source of spatial information (AET).
It seems at the end that this second part is not really a calibration framework, since you even introduce some modifications in the model structure to properly incorporate spatial information in modelled AET. It seems to me that the main outcome of the paper is an interesting way to integrate spatially distributed observations in a modeling framework. It is a kind of spatial data assimilation, or better spatial data integration approach, more than a traditional model´s parameters calibration. I suggest in the Introduction and in the Discussion to place your results in the broader perspective of the data integration techniques and not only model calibration literature, and underline this potential of your proposed approach.

**Specific comments:**

**Introduction**

It is important to underline that actual evapotranspiration (AET) estimates from satellites are not observations, but the results of a model (in this case the TSEB). This model shares part of the input information used by the hydrological model (i.e. LAI, fractional vegetation cover, meteorological data as temperature). This does not influence paper´s results, but it is important to clearly state that AET patterns are not observations, but the results of a remote-sensing based model, with a lot of uncertainties.

**2.2 Satellite based data**

Canopy fraction is a key input information of the TSEB model. How it has been estimated?

AET estimations of the TSEB model are relative to the instant of the thermal remote sensing image used. How AET estimations have been extended to the daily and monthly time scale?

**Figure 1** TSEB results are usually highly dependent on LST and canopy fraction maps. It is very interesting how the RS-AET map reflects also well the soil type map, since no soil or soil water content information is given to the TSEB. Your hypothesis to link soil type to root density and therefore to root

water extraction is interesting. However other hypotheses could be done. The simplest is just the areal with lower sand content are wetter, because the soil has a higher water holding capacity, and therefore there is a higher AET.  Could you comment on this?

**Distributed root fraction coefficient**. The assumed model could be better described. Why land cover does not influence root fraction? I expect that different vegetation types are the main source of the different root fractions. Is the root depth a relevant parameter in the mHM model or only root fraction is used?

**4.1 Objective functions.** The application of the new SPAEF index is one of the main elements of novelty in the paper. A schematic figure explaining the concept of "histogram intersection" could help. It is not obvious for me.

**Figure 3.** Why including AET patterns reduces model performances in the summer months? This is the time when I expect AET counts more for the water budget. Is this a drawback of the choice of not considering biases in AET? What would happen if you include also as calibration target the temporal evolution of the spatial average of AET?

**Figure 4.** This figure shows the potential of this technique of spatial calibration more in terms of remote sensing data integration, than in terms of calibration.  In fact, in many cases distributed hydrological models cannot produce detailed spatial patterns, because the coarse spatial patterns of models input data (except topography, soil and land cover types are usually known at a coarse spatial scale). Could you comment in this? (see also general comments)

**Page 16** Results show how combining Q and AET results in a more robust model parametrization for the validation phase. Could this also contribute to reduce models equifinality?

**Page 16 "***It is recognized that traditional downstream discharge measurements contain much more accurate and robust information on the overall water balance compared to the non-continuous remotely sensed estimates***". Please provide some references to support this.

**Discussion**

**Page 17 "***Here, we ignore the temporal aspect and focus only on the consistent spatial patterns for each month of the growing season***" "***The current calibration framework builds on the assumption that the satellite based estimate of AET patterns approximate an observed patterns that is suitable for model optimization***". Please motivates better this assumption. See general comments above.

The choice of not including temporal information from the AET data is an assumption quite well justified in the discussion.  However, this could reduce the information amount coming from AET. "*AET estimate is validated against eddy covariance stations (Mendiguren et al., 2017) they only represent specific cloud-free days limiting their value to assess the long term water balance of the catchment*" OK, but if AET patters in cloudy days are different? What happens if AET patterns are not stationary in time?  From my experience with distributed models, looking to daily model´s AET output maps, I´ve been ever surprised how such patterns are changing from day to day. See also general comments. A preliminary analysis of the time series of the RS-AET map should be done to assess how much time-consistent are the patterns.

---

## Author Comment (AC1) · 27 Dec 2017

**Reply to Reviewer Comments**

**Reviewer #1**

Comment 1) This paper deals with an innovative calibration framework that combines temporally aggregated observed spatial patterns with a new spatial performance metric and a flexible spatial parameterisation scheme. An application of the mesoscale Hydrologic Model (mHM) to the Skjern River Basin is used as an example show the effectiveness of the presented calibration framework. This is a very timely topic that fits very well to the scope of HESS.

However, the spatial model parameterizing methodology is not well described. Both the root fraction coefficient and the PET correction factor parameterizations should be presented in more detail including graphical presentations of the underlying relationships. For these reasons, my recommendation is to accept this manuscript with minor revisions.

I have provided specific comments and suggestions for improvement below.

*Reply from authors: The authors thank the Dr Heye Bogena (reviewer #1) for his constructive comments on the manuscript. We have replied to each comment below.*

**Specific comments:**

Comment 2) P3L7: Which models?

*Reply from authors: "Immerzeel and Droogers (2008) showed that the models can be constrained by using spatially distributed observations with a monthly temporal resolution" is replaced by "Immerzeel and Droogers (2008) showed how a semi distributed model of a basin in Southern India could be constrained by using spatially distributed observations with a monthly temporal resolution"*

Comment 3) P3L16-32: This section is redundant with the method section and should be shortened.

*Reply from authors: We agree with the comment. This section will be shortened in the revised version of the manuscript.*

Comment 4) P5L6 Delete "s"

*Reply from authors: "s" is removed.*

Comment 5) P5L9-11: How accurate are the AET maps, e.g. in relation to continuous EC-measurements?

*Reply from authors: The accuracy of the AET maps has not been quantified in this paper. However, a comparison to observed AET for three EC sites has been included in a previous paper Mendiguren et al 2017 HESS. This comparison focussed on the monthly mean AET estimates for three land-cover types, which were well in agreement with the measurements, although it has to be noted that the measurements themselves were subject to uncertainty due to energy balance closure issues. Below is the figure that appears in the Mendiguren et al. 2017 paper*

[Figure]

*However, in the current study, the accuracy of the AET maps cannot be completely assessed by a comparison to EC measurements, because only the bias insensitive pattern information is utilized, and the uncertainty of this pattern cannot be fully described by just three EC sites. The*

*current study therefore relies on an assumption that when performing well for monthly means at three different sites and being mainly driven by remote sensing observations of LST and NDVI/LAI, the satellite based AET estimates contain spatial pattern information that is suitable for constraining the spatial pattern simulations of our distributed model.*

Comment 6) P5L30-31: Why should this procedure accelerate model runs?

*Reply from authors: The file size of the 1x1km inputs is much larger than 10 or 20 km. Therefore, reading these files to the model at each timestep takes very long time as compared to the discretisation algorithm defined inside the Fortran code. This is a technicality related to either pre-processing the LAI-based input on a daily scale prior to model execution or reading the monthly data and disaggregate to daily data inside mHM. We suggest skipping this line altogether.*

Comment 7) P6L8: "stretch the spatial contrast" of what?

*Reply from authors: Corrected as "spatial contrast of simulated actual evapotranspiration" in the revised version of the manuscript.*

Comment 8) P6L8: ": : :based on soil and vegetation properties: : :"

*Reply from authors: Corrected.*

Comment 9) P6L11: Please change "domain-specific" into "site-specific" or "local"

*Reply from authors: Corrected throughout the manuscript.*

Comment 10) P6L11-12: According Feddes et al. (2001) and others, the root fraction coefficient is a vegetation dependent coefficient. Please add more information on why your assumption that this coefficient can be explained by field capacity is justified.

*Reply from authors: It is true that generally the literature suggests that root fraction coefficient is mainly dependant on vegetation type. We also separate root fraction coefficient in two main categories: forest and non-forest. However, for the non-forest (in this study agricultural cropland, since this is the only major land cover type) we have introduced an option to distribute root fraction coefficient based on soil type. This is based on several studies carried out by Danish soil scientists who showed that for the very sandy soils of Western Denmark the effective root depth is smaller relative to soils with similar crop types, but higher clay content, although this dependency seizes at a certain clay content. Instead of basing the root fraction*

*coefficient on soil type, we utilized the FC which in mHM is a function of sand and clay content through the internal pedo transfer functions. We admit that the relation might be specific to the Danish case, where even very poor (very sandy) soils are utilized for crop production and where the relation between root depth and soil type has been established for very sandy to loamy soils. A similar relation is used in the parametrisation of the Danish National Water Resources model and in National crop growth models. In the revised paper we will elaborate on this and make it clear that root depth is also a function of main landcover/vegetation types.*

Comment 11) P6L14-15: At this point it unclear why these parameterizations increase the model freedom. Please reorganise the text in a more comprehensively way.

*Reply from authors: We will rephrase and organise this paragraph more thoroughly in the revised version of the manuscript.*

Comment 12) P6L19-24: It took me some time to understand your method, also because the assumption that the root fraction is linked to FC is counterintuitive. For a better understanding of your method, it would be helpful to graphically show the relationship between FC and rooting depth based on the soil database.

*Reply from authors: We will elaborate on this, but the relation builds on previous work by others, so we can refer to their studies and illustrate the relation graphically but not plot their actual data.*

Comment 13) P6L22: Why do you restrict your method to pervious non-forested areas?

*Reply from authors: Basically mHM operates with three land cover types: Forest, pervious and impervious (urban). The urban areas in our catchment is very limited, so we ignore that category, leaving just forest and non-forest. As explained above we acknowledge that the overall vegetation type (forest or cropland) is the dominating controlling factor for root depth, therefore we allow separate root fraction coefficients for forest and non-forest and only apply the FC dependency for non-forest.*

Comment 14) P6L22: Please change "domain-specific" into "site-specific" or "local"

*Reply from authors: Corrected.*

Comment 15) P7L2: Why should soil properties of sandy soils impede root development?

*Reply from authors: Impede is the word used in the literature we refer to on the root depth dependency on soil type.*

Comment 16) P7L2-3: Which parameters and why is a fine vertical discretization more effective?

*Reply from authors: This is related to the way the root fraction coefficient effects AET simulations. When a given soil layer dries out (SM falls below FC) the simulated AET is reduced linearly until it reaches zero at wilting point. Fine vertical discretization of the upper soil layers will allow for a finer representation of the root fraction variation with depth and result in more frequent occasions where SM content reduces AET. This might be a technicality, and the sentence could be omitted, since we used the same vertical discretization through all model runs in this study and we have not examined the impact of vertical discretization.*

Comment 17) P7L5-7: By changing the root fraction parameters for maximum and minimum FC, the relationship between FC and rooting depth will be changed. In the extreme case, both parameters have the same value, which means that there is no relationship at all. Did you check whether the optimised model still provides realistic relationships between FC and rooting depth? In addition, it is unclear, how you derived FC values from your digital soil map. Typically, one would sum up the horizontal FC values down to the certain soil depth (e.g. rooting depth). Please provide additional information.

*Reply from authors: For the latter question, The FC is estimated in the pre-processing within mHM based on pedo transfer functions. It is a fundamental part of the model concept that only soil texture data is specified and the soil physical properties are derived. This insures a spatially consistent parametrization (given the soil texture data is good) and reduces the calibration parameters to the global pedo transfer functions parameters. We used the same soil texture parametrization (and therefore FC) for the entire soil column. We have looked at the variation in both the derived FC and WP maps and the range of root fraction parameters and they all look reasonable. In addition, we have tied the minimum and maximum root fraction coefficient so they can approach each other but never reverse. If they approach each other through calibration and become identical, that would essentially indicate that the approach did not benefit the spatial pattern performance, but that is not what we see. It clearly improves the spatial pattern performance to separate root fractions based on FC.*

Comment 18) P7L16-17: You should not mention equations that have not been already introduced.

*Reply from authors: Corrected.*

Comment 19) P7L21-25: The section needs to be rewritten in a more comprehensible way.

*Reply from authors: We will reorganise this section in the revised version of the manuscript.*

Comment 20) P7L26: It is how this equation was derived and why it should be "physically meaningful". In addition, it is unclear how the DSF parameter is used to correct ET_ref.

*Reply from authors: We didn't derive this equation. It is a time-space variable implementation of the crop coefficient concept suggested by Allen et.al. The basis is that the climate dependant reference evapotranspiration is given at a coarse scale of 20 km grid for a reference crop. In reality, the vegetation does not reflect a reference crop everywhere. In order to correct the reference ET a scaling factor is applied which is above 1 if the evaporative potential of the vegetation is higher than for the reference crop and below 1 if it is lower. Typical values of crop coefficients are between 0.8 and 1.2. Allen et al. 1998 and others (eg. Hunink et al. (2017)) suggested using LAI or NDVI to estimate the crop coefficient.*

*Our implementation is simply using the same equation in combination with remote sensing based LAI to create a time-space variable correction factor to convert ETref to ETpot.*

*The DFS is simply a multiplication factor. ETpot = DFS\*ETref*

*mHM originally contains an even simpler correction factor, which is spatially and temporally uniform (although it also has an option to include aspect in mountainous terrain). We have omitted this correction and implicitly included it in the calibration of the DFS, because the average DFS does not necessarily add up to 1. We will be more accurate in the description of the DFS in the revised manuscript.*

Comment 21) P7L28: The previous sections also belong to "Methods".

*Reply from authors: We don't regard the parametrization of the mHM model as a methodology as such, we are not changing the process descriptions of the model but merely adding spatial flexibility to the parametrization of exiting model parameters. Therefore we feel that it is more appropriate to limit the methodology section to the more novel parts of the manuscript, namely the development of a new spatial performance metric and a multi-objective calibration framework based on a complimentary principle.*

Comment 22) P9L1: Only accepted paper should be used a reference.

*Reply from authors: We agree with the comment.*

Comment 23) P9L4-9: This section can be omitted.

*Reply from authors: will be corrected*

Comment 24) P9L24-26: Is this statement relevant for this work or Koch et al., 2017a

*Reply from authors: This statement is relevant only for Koch et al (2017) and will be omitted.*

Comment 25) P9L34: Why are you using the "same" cloud-free days?

*Reply from authors: This perhaps should be rephrased; we want to explain that we are making the monthly average AET maps based on simulated daily AET by averaging only the days that were also available for estimating the RS based maps (cloud free days).*

Comment 26) P10L5: Either use the Greek symbol for phi or "phi" (here and elsewhere)

*Reply from authors: Corrected.*

Comment 27) P10L22: Delete "and"

*Reply from authors: Corrected.*

Comment 28) P10L33-34: I wonder why the parameter "root fraction" for forest area is listed and not for the impervious non-forest areas, since the latter was used for the spatial parameterization.

*Reply from authors: Impervious areas are very small and not represented in the land use map for root fraction distribution. Therefore, root fraction for impervious areas is not listed in Table 2.*

Comment 29) P10L34: "(SPAEF column in Tab. 2)"

*Reply from authors: Corrected.*

Comment 30) P11L9-10: Why are you using a second model warm-up period (2005-2008)? This is also not mentioned in the text.

*Reply from authors: The second warm-up period is just used to ensure that the validation results are not effected by initial conditions. The text will be updated accordingly in the revised version of the manuscript.*

Comment 31) P12L18-20: However, it should be noted that the uncertainty of the spatial AET information from remote sensing is typically larger than the runoff measurements.

*Reply from authors: We completely agree that the average AET estimate from RS is more uncertain that the runoff measurement. That is the reason why we use only the runoff to constrain the water balance, while we only use the bias insensitive pattern information of the*

*RS AET to constrain the simulated pattern. It is unclear to us exactly what to correct or rephrase here.*

Comment 32) P12L28: Delete "that"

*Reply from authors: Corrected.*

Comment 33) P13L2-3: This indicates that the spatial AET information either has large uncertainties or that it has very limitated information on the subsurface properties due to extensive irrigation in the catchment. Please discuss.

*Reply from authors: We disagree, the poor performance on discharge for the spatial-only calibration is a direct consequence of the way the SPAEF objective function is designed. Since it is bias-insensitive and only contains information on the spatial pattern it cannot be used to constrain the discharge and water balance. The Spatial only calibration is (as stated in the manuscript) not meaningful from a water balance or discharge perspective, it is solely included as a benchmark for the spatial pattern performance capability of the modelling framework and to illustrate the limited trade-offs between discharge and SPAEF when applied as proposed in the current study.*

Comment 34) P15L13: Does Figure 4 present a certain year or monthly averages of several years?

*Reply from authors: As mentioned in Page 5 line 6-10, Figure 4 presents averages of cloud free days for a specific month across all years for the model calibration period (2001-2008).*

Comment 35) P15L14: The differences seem to vary also from month to month. You should quantify the differences in AET patters, e.g. by presenting the variances or variograms.

*Reply from authors: What was meant by P15L14 was simply that the calibrations including the RS AET maps result in simulated patterns that has a much better representation of the variance compared to the Q-only calibration. Table 3 and 4 present month to month differences in SPAEF which also includes the coefficient of variation. It is unclear what a specific presentation of variances across months would add to the analysis, we are open to hear more on this from the reviewer.*

Comment 36) P17L21: Awkward sentence. Please reformulate.

*Reply from authors: Corrected as below.*

*"This is largely attributed to the nature of the metric as the spatial performance metric is bias-insensitive whereas the streamflow metrics have very little sensitivity to spatial redistribution of AET patterns as long as the spatial averages remain unchanged."*

Comment 37) P17L25: "This was: : :"

*Reply from authors: Corrected as below.*

*"This was done because even though the satellite based AET estimate is validated against eddy covariance stations (Mendiguren et al., 2017) they only represent specific cloud-free days limiting their value to assess the long term water balance of the catchment."*

Comment 38) P617L31-32: Unclear why the sandy soil texture should restrict the model parametrisation method.

*Reply from authors: What is meant here is that the root fraction coefficient dependency on soil type might be site-specific due to the uniform land use (agricultural cropland) across soils ranging from very coarse sandy soil to loamier soils. In other words, we do not claim that this parametrisation is generic, but it reflects knowledge of the particular area and fits well with the observed patterns in soil properties and RS AET as illustrated in Figure 1. We will rephrase this sentence in a revised manuscript.*

Comment 39) P617L32: Please change "domain-specific" into "site-specific" or "local"

*Reply from authors: Corrected.*

Comment 40) P18L23: "has proven"

*Reply from authors: Corrected.*

**Figures and Tables:**

Comment 41) Figure 1: The RS-AET scale should be reversed (red should indicate high AET values)

*Reply from authors: We disagree, we have used green/blue colours for high AET and red/brown for low throughout the manuscript. To us this is the most logical colouring scheme, which intuitively symbolizes wet and dry conditions.*

Comment 42) Table 1: Columns 3, 4 and 5 should be removed as they provide only limited information, which is already presented in the text.

*Reply from authors: Corrected.*

Additional literature

Feddes, R.A., H. Hoff, M. Bruen, T. Dawson, P. de Rosnay, P. Dirmeyer, R.B. Jackson, P. Kabat, A. Kleidon, A. Lilly, and A.J. Pitman, 2001: Modeling Root Water Uptake in Hydrological and Climate Models. Bull. Amer. Meteor. Soc., 82, 2797–2809.

---

## Author Comment (AC2) · 27 Dec 2017

**Reply to Reviewer Comments**

**Reviewer #2**

The authors present a methodology aimed to improve predictions of a distributed hydrological model both in time and space. In order to identify model parameters responsible for the spatial predictions, an additional complex objective function is introduced taking into account the match between observed (satellite based) spatial evapotranspiration patterns and those predicted by the model.

The paper is interesting and the approach taken opens a new direction of research towards the use of spatial remote sensed observations to improve predictions of a distributed hydrological model. In particular, the authors applied site-specific parameterizations to increase the flexibility of the description of actual evapotranspiration characteristics (root zone and potential evapotranspiration corrections). The authors formulated the problem in a deterministic framework, which might be a good introduction to a new approach, but the discussion on uncertainty is missing. The paper is well written and requires only some clarification of the presented material and a wider discussion of the assumptions taken. I recommend to publish the paper after minor corrections.

*Reply from authors: The authors thank the Dr Renata Romanowicz (reviewer #2) for her positive and constructive comments on the manuscript.*

**Specific comments**

Comment 43) Page 1, line 18    In addition two new site-specific spatial parameter distribution options have been introduced

*Reply from authors: Corrected.*

Comment 44) Page 2, line 19 This is from the fact    grammar should be corrected e.g. 'this is because..'

*Reply from authors: Corrected.*

Comment 45) Page 3, Line 17 it is not clear what the authors mean by 'domain'

*Reply from authors: We meant "catchment" with domain. This is corrected in the revised version of the manuscript.*

Comment 46) Page 3, line 22 is it    for comparing spatial patterns of two continuous variables?

*Reply from authors: Corrected.*

Comment 47) Page 5, lines 8-11 what is the uncertainty of AET estimates?

*Reply from authors: Please see reply to comment 5) by the reviewer #1*

Comment 48) Page 5, line 30 Could you please give more detail about the way monthly AET maps are applied in the model and the disaggregation method used?

*Reply from authors: We will elaborate on this in the revised manuscript.*

Comment 49) Page 6, line 9 The parameterisation introduced is a very interesting way forward and requires a separate paper backed up with field experiments. Could you please give the possible disadvantages of the parameterisation? Even though the parameterisation decreases the number of parameters of a distributed hydrological model, the parameters require a sufficient amount of observations to be properly identified. The question is how to test the parameterisation using very limited and uncertain information obtained from the indirect and fragmented satellite observations. The other question is, how to estimate the uncertainty related to that parameterisation. Some comments would be welcome.

*Reply from authors: Our study builds on previous studies on the relations between surface properties of soil and vegetation and model parameters. We are not deriving these relationships ourselves from new field data. We don't see the spatial parametrization as a novel methodology, but more as a flexible spatial parametrization scheme, that allows us to explore the main topic of the paper, which is the use of a new spatial pattern metric and complementary observations of RS AET patterns and stream discharge for model calibration. The idea behind the parametrization is that we use data that is well described spatially, namely soil texture maps (based on a great number of measurements) and vegetation maps (based on satellite remote sensing) to distribute related model parameters spatially while still allowing for some calibration. The disadvantage would be the validity of the proposed relations between soil and vegetation properties and model parameters. But we regard it as a robust approach that avoids over simplification in spatially uniform parametrization and over-parametrization in grid-by-grid calibration.*

Comment 50) Page 6, lines 10-24 How sensitive is water storage variability to this parameterisation?

*Reply from authors: We don't quite understand the question, perhaps the reviewer could elaborate.*

Comment 51) Page 7, lines 2-10 was the model tested on observations and what assumptions must be fulfilled?

*Reply from authors: This concept is taken from Danish soil type and root depth studies (see references in manuscript) and is believed to be representative for the study area, where cereal crops are grown on soils ranging from very sandy to loamy. See also reply to reviewer 1.*

Comment 52) Page 7, line 24 was this parameterisation tested on observations? What assumptions are imposed?

*Reply from authors: Please see reply to comment 20) by reviewer #1.*

Comment 53) Page 8, line 7 Since comparison should start from a new line and a new objective function responsible for reproducing spatial patterns should be introduced.

*Reply from authors: We agree, this will be corrected.*

Comment 54) Page 8, lines 10-11 histograms of what?

*Reply from authors: Corrected as below. Also we will elaborate on the histogram matching as requested by reviewer #3.*

*"In this context, we adopted the structure of the Kling–Gupta efficiency while substituting the standard deviation term by a term based on the coefficient of variation $\left(\sigma_O/\sigma_S\right)$ and replacing the bias term with a histogram comparison index to compare the intersection-percentage of two histograms of observed and simulated spatial maps."*

Comment 55) Page 8, line 24 The AET from TSEB have been treated as error free data – a comment is needed on the possible errors involved.

*Reply from authors: This is a very good point, and we have given it a lot of thought how to quantify the uncertainty. However, given the way we utilize the AET maps (bias-insensitive pattern performance) the quantitative uncertainty related to comparison with point measurements would not be very useful. What is needed is a quantification of the uncertainty of the RS AET pattern. This is far from trivial, because even if the uncertainty of some of the input to the TSEB model (mainly LST) can be approximated, it is the impact this uncertainty has on the AET pattern that is important. The crude assumption is that errors in the TSEB input are largely uniform for this relatively small catchment (e.g. if the LST input on a given day has an estimated error of +1 Kelvin, this is assumed to apply for the entire area and therefore have a limited effect on the estimated AET pattern. The whole framework is deterministic, and this*

*is clearly a limitation, however, the study proposes several new ideas, and to put this into a thorough uncertainty framework is beyond the scope, but will be an interesting further development.*

Comment 56) Page 9, line 31 That criterion might be very misleading when the response surface is flat. The optimisation algorithm might stop in any part of the optimisation range or, most likely at the edge of the parameter range. The authors are asked for a comment.

*Reply from authors: Each iteration is comprised of more than 100 runs with different parameter sets. This means that if the objective function doesn't improve after more than 500 runs then the calibration stops. This is assumed to be a reasonable number of testing in an optimization and generally we don't see parameters that stop at the edge of the parameter range. We will add more details about iterations in the revised version of the manuscript.*

Comment 57) Page 10, line 8   A and B and   . (- B is confusing)

*Reply from authors: Corrected.*

Comment 58) Page 13, line 12 The existence of local minima depends on the form of the objective function which defines the parameter response surface. In the case of a model with 26 parameters the objective function will show local minima. Following the equifinality hypothesis, there are many parameter sets which give the same value of the objective function and therefore it is not surprising that many different optimum solutions can be found.

*Reply from authors: We agree, we will revise this sentence, it actually does not raise questions about the SCE, but is a result of the multi-objective 26 parameter problem being solved.*

Comment 59) Page 14, line 8-9 Spatial calibration constraints the solution rather than reduces its uncertainty – the uncertainty was not evaluated.

*Reply from authors: Correct, we will rephrase this from reduces uncertainty to better constrained.*

Comment 60) Page 15, lines 23-24 Could it be explained why the improvement occurs?

*Reply from authors: We describe a drop in performance, so I guess the question is why the performance drops? And yes we have a good idea why; there was a significant drop in the number of rain gauges in Denmark in 2007-onwards, this has had a direct impact quality of the precipitation input and on the streamflow performance across the country, but we fell it is beyond the scope of this paper to go into that discussion. It is the topic of other ongoing studies.*

Comment 61) Page 16, Discussion As I understood, the authors did calibration using different sequences of cases. Might it be advantageous if the optimisations Q-only and Spatial only were applied iteratively?

*Reply from authors: Yes, we thought about that and given the limited trade-off and that some model parameters are mainly/only effecting either discharge or SPAEF, we would probably have reached similar results as the combined calibration.*

Comment 62) Page 17, line 13   associated with   .

*Reply from authors: Corrected.*

Comment 63) Page 17, line 32    site-specific due to   .

*Reply from authors: Corrected.*

Comment 64) Page 17, line 34   different countries.

*Reply from authors: Corrected.*

Comment 65) Page 18, line 28 The conclusion on achieving a more robust parameter set because the trade-off disappears is not well founded and too general after only one validation exercise.

*Reply from authors: We will rephrase this, not claiming that the parameter set is more robust.*

---

## Author Comment (AC3) · 27 Dec 2017

**Reply to Reviewer Comments**

**Reviewer #3**

Review of the Article

Combining satellite data and appropriate objective functions for improved spatial pattern performance of a distributed hydrologic model by Mehmet C. Demirel, Juliane Mai, Gorka Mendiguren, Julian Koch, Luis Samaniego, Simon Stisen

The paper addresses the relevant topic of using spatial data for the calibration of distributed hydrological models. The paper explores the tradeoffs between streamflow-based and spatial-based calibrations, illustrating the benefits of combining separate observation types and objective functions. Given the increasing availability of spatial data, to understand the value of a spatial calibration versus a more traditional calibration is a timing and relevant research topic. Spatial pattern of actual evapotranspiration (AET), obtained with the mesoscale Hydrologic Model (mHM), are calibrated against spatial patterns of AET estimated trough remote sensing (TSEB model). The major novel point of the paper is to present novel bias-insensitive spatial pattern metric, which exploits the key information contained in the observed patterns, useful for spatially distributed models optimization. I found the paper valuable and worth to be published. However, I believe some methodological aspects should be further clarified, as detailed in the specific comments.

I suggest a minor revision for the paper. Please revise also carefully English language.

*Reply from authors: We wish to thank Dr. Giacomo Bertoldi (Reviewer #3) for the very constructive comments and suggestions.*

General comments:

**1.**In the paper formulates the hypothesis that "The current calibration framework builds on the assumption that the satellite based estimate of AET patterns approximate an observed pattern that is suitable for model optimization". In my advice, this hypothesis should be better motivated and the implications better discussed.

At the end, six monthly mean "climatological" AET maps have been produced. But spatial patterns are time consistent? If the spatial structure of AET changes a lot from time to time, or there is a great inter-annual variability, then it makes little sense to calibrate the model to such patterns. A preliminary analysis on the temporal variability of the observed AET patterns should be done. Only if spatial AET patterns are quite constant over the different years, the proposed calibration approach could be used. Moreover, the fact that an AET monthly "climatology" has been used, could be one reason of the limited tradeoff found with the discharge. Could you discuss about this? In this way,

you cannot for example identify from AET observation a drought period, which can have an effect on discharge.

I see later that this choice is explicit in the paper (see 4.1 Objective functions). "Since all three terms are bias-insensitive, the spatial efficiency only constrains the model simulations with the pattern information in the satellite data while leaving the water balance (bias) to be constrained by streamflow". However, this implies a hypothesis of spatial invariance of AET patterns. You need to support and discuss this assumption in the paper.

*Reply from authors: We will elaborate on the assumption that the RS based AET data contains valuable pattern information that can help constrain the hydrological model regarding simulation of spatial patterns. We did a preliminary analysis of how consistent the spatial patterns were in time. They turned out to be quite consistent, this we will include in the final reply and potentially add to the manuscript as either a discussion or added figure/table. The overall idea is to utilize the most informative part of the observation data (spatial patterns from RS AET maps and water balance and temporal pattern from discharge). In addition, we are aiming at calibrating mainly the spatial parametrization of soil and vegetation parameters using the bias-insensitive spatial pattern metric, these parameters typically do not vary in time (expect in some cases vegetation, like in our case vegetation dynamics change, but the vegetation related calibration parameters are time-invariant). Therefore, we believe that the temporally averaged spatial patterns are a reasonable way of constraining the spatial model parametrization. In addition, a spatially consistent drought would be captured by the discharge observations; however, admittedly our approach is not designed to capture spatial variability in drought patterns (this would especially be relevant for larger catchments).*

**2.** Another assumption is that "Bias and temporal variability of satellite derived AET estimates could be useful for model optimization, however, in this study, we deliberately limited the information content of the satellite data to address the spatial patterns." You clearly separate in your calibration framework the source of temporal information (discharge) from the source of spatial information (AET). It seems at the end that this second part is not really a calibration framework, since you even introduce some modifications in the model structure to properly incorporate spatial information in modelled AET. It seems to me that the main outcome of the paper is an interesting way to integrate spatially distributed observations in a modeling framework. It is a kind of spatial data assimilation, or better spatial data integration approach, more than a traditional model´s parameters calibration. I suggest in the Introduction and in the Discussion to place your results in the broader perspective of the data integration techniques and not only model calibration literature, and underline this potential of your proposed approach.

*Reply from authors: I see the point; however, I still regard this as a classical model parameter calibration approach different from data assimilation schemes. The reason is that we do not update states, fluxes or parameters directly to match the observed pattern, but only use the observed spatial patterns to formulate a classical objective function that is then minimized by changing global parameter values. Some of these global parameters are then part of an upscaling operator or transfer function, but they are essentially still model parameters. In the revised manuscript, we will make a distinction between our approach and data assimilation.*

**Specific comments:**

Introduction

It is important to underline that actual evapotranspiration (AET) estimates from satellites are not observations, but the results of a model (in this case the TSEB). This model shares part of the input information used by the hydrological model (i.e. LAI, fractional vegetation cover, meteorological data as temperature). This does not influence paper´s results, but it is important to clearly state that AET patterns are not observations, but the results of a remote-sensing based model, with a lot of uncertainties.

*Reply from authors: True, this should be made very clear in the revised manuscript.*

2.2 Satellite based data

Canopy fraction is a key input information of the TSEB model. How it has been estimated? AET estimations of the TSEB model are relative to the instant of the thermal remote sensing image used. How AET estimations have been extended to the daily and monthly time scale?

*Reply from authors: Details of this are given in: Mendiguren, G., Koch, J., and Stisen, S.: Spatial pattern evaluation of a calibrated national hydrological model – a remote-sensing-based diagnostic approach, Hydrol. Earth Syst. Sci., 21, 5987-6005, https://doi.org/10.5194/hess-21-5987-2017, 2017. We will make sure reference is given in the manuscript.*

Figure 1 TSEB results are usually highly dependent on LST and canopy fraction maps. It is very interesting how the RS-AET map reflects also well the soil type map, since no soil or soil water content information is given to the TSEB. Your hypothesis to link soil type to root density and therefore to root water extraction is interesting. However other hypotheses could be done. The simplest is just the areal with lower sand content are wetter, because the soil has a higher water holding capacity, and therefore there is a higher AET. Could you comment on this?

*Reply from authors: Yes, our initial thought was that simply soil physical properties could explain the differences. However, this was not enough to explain the differences seen in the RS AET maps. In addition, we had specific knowledge from the literature about the root density variations in this region. Also the sensitivity of root density to AET is very high in the model (and in many other PET based AET models as well), so getting the soil influence on AET right while preserving the original uniform root density for all non-forest areas would not be sufficient or correct.*

Distributed root fraction coefficient. The assumed model could be better described. Why land cover does not influence root fraction? I expect that different vegetation types are the main source of the different root fractions. Is the root depth a relevant parameter in the mHM model or only root fraction is used?

*Reply from authors: mHM uses root fraction distribution, and land cover is the main driver for differences in root fraction, but it is simplified to forest, non-forest vegetation and impervious (the later not being used in the current study due to very minor area of urbanization). The soil dependency we introduce is only to further distribute the root density across non-forest vegetation (agricultural cropland). The main differences in root density is still between forest and non-forest. This will be made a bit clearer in the manuscript.*

4.1 Objective functions. The application of the new SPAEF index is one of the main elements of novelty in the paper. A schematic figure explaining the concept of "histogram intersection" could help. It is not obvious for me.

*Reply from authors: The histogram intersect will be explained better.*

Figure 3. Why including AET patterns reduces model performances in the summer months? This is the time when I expect AET counts more for the water budget. Is this a drawback of the choice of not considering biases in AET? What would happen if you include also as calibration target the temporal evolution of the spatial average of AET?

*Reply from authors: This is probably a drawback, and the loss in performance in discharge during summer months is seen as a tradeoff resulting from adding another objective function.*

Figure 4. This figure shows the potential of this technique of spatial calibration more in terms of remote sensing data integration, than in terms of calibration. In fact, in many cases distributed hydrological models cannot produce detailed spatial patterns, because the coarse spatial patterns of models input data (except topography, soil and land cover types are usually known at a coarse spatial scale). Could you comment in this? (see also general comments)

*Reply from authors: As replied above, we regard this as a classical parameter calibration exercise. However, the flexibility in the spatial model parametrization and the use of distributed input (with*

*same resolutions as the model grid) of soil texture and vegetation enables the model to approach a detailed spatial pattern. It should be noted that the results of the Q-only calibration in figure 4 has exactly the same parametrization scheme and free model parameters as the two other calibrations, so the only reason that it fails in reproducing the spatial pattern is that it is not informed about what the "true" pattern is.*

Page 16 Results show how combining Q and AET results in a more robust model parametrization for the validation phase. Could this also contribute to reduce models equifinality?

*Reply from authors: We hope so; however, we have not addressed equifinality as such.*

Page 16 "It is recognized that traditional downstream discharge measurements contain much more accurate and robust information on the overall water balance compared to the non-continuous remotely sensed estimates". Please provide some references to support this.

*Reply from authors: Not sure there are specific references for this quite specific statement. We are here arguing that a continuous time-series of total catchment discharge contains more information on the overall water balance compared to a series instantaneous snapshots of AET for only cloud free conditions. We will rephrase to make this our own statement, unless we find an appropriate reference.*

Discussion

Page 17 "Here, we ignore the temporal aspect and focus only on the consistent spatial patterns for each month of the growing season" "The current calibration framework builds on the assumption that the satellite based estimate of AET patterns approximate an observed patterns that is suitable for model optimization". Please motivates better this assumption. See general comments above.

*Reply from authors: We will motivate this better in the revised manuscript.*

The choice of not including temporal information from the AET data is an assumption quite well justified in the discussion. However, this could reduce the information amount coming from AET.

*Reply from authors: true, we could miss information actually contained in the AET data, however, as explained throughout the manuscript, we focus on the information content of the RS AET that supplements the discharge time series the best.*

"AET estimate is validated against eddy covariance stations (Mendiguren et al., 2017) they only represent specific cloud-free days limiting their value to assess the long term water balance of the catchment" OK, but if AET patters in cloudy days are different? What happens if AET patterns are not stationary in time? From my experience with distributed models, looking to daily model´s AET output maps, I´ve been ever surprised how such patterns are changing from day to day. See also general comments. A preliminary analysis of the time series of the RS-AET map should be done to assess how much time-consistent are the patterns.

*Reply from authors: This is definitely a limitation to our approach. But again, we have decided to utilize what we assume is the best information content available from the AET data. This is the spatial pattern on the days where we have LST observations. We could have interpolated these cloud-free "observations" to cloudy and rainy days, however this would not increase our number of "true" observations. In addition, for the current catchment, the impact of soil and vegetation parameters (which we calibrate) on AET on very cloudy and rainy days will be very limited, because in this region, potential Evapotranspiration will occur as soon as the potential evapotranspiration becomes low (as will happen on cloudy and rainy days).*